# TRANSFORMER-PATCHER:
# ONE MISTAKE WORTH ONE NEURON

**Zeyu Huang**[1,2]**, Yikang Shen**[4]**, Xiaofeng Zhang**[1,2]**, Jie Zhou**[5]**, Wenge Rong**[1,3]**, Zhang Xiong**[1,3]

[1]State Key Laboratory of Software Development Environment, Beihang University, China
[2]Sino-French Engineer School, Beihang University, China
[3]School of Computer Science and Engineering, Beihang University, China
[4]Mila, University of Montreal, Canada, [5]WeChat AI, Tencent Inc, China
{zeroy.huang,yikang.shn}@gmail.com,withtomzhou@tencent.com
{xiaofeng_z,w.rong,xiongz}@buaa.edu.cn

## ABSTRACT

Large Transformer-based Pretrained Language Models (PLMs) dominate almost all Natural Language Processing (NLP) tasks. Nevertheless, they still make mistakes from time to time. For a model deployed in an industrial environment, fixing these mistakes quickly and robustly is vital to improve user experiences. Previous works formalize such problems as Model Editing (ME) and mostly focus on fixing one mistake. However, the one-mistake-fixing scenario is not an accurate abstraction of the real-world challenge. In the deployment of AI services, there are ever-emerging mistakes, and the same mistake may recur if not corrected in time. Thus a preferable solution is to rectify the mistakes as soon as they appear nonstop. Therefore, we extend the existing ME into Sequential Model Editing (SME) to help develop more practical editing methods. Our study shows that most current ME methods could yield unsatisfying results in this scenario. We then introduce Transformer-Patcher, a novel model editor that can shift the behavior of transformer-based models by simply adding and training a few neurons in the last Feed-Forward Network layer. Experimental results on both classification and generation tasks show that Transformer-Patcher can successively correct up to thousands of errors (*Reliability*) and generalize to their equivalent inputs (*Generality*) while retaining the model's accuracy on irrelevant inputs (*Locality*). Our method outperforms previous fine-tuning and HyperNetwork-based methods and achieves state-of-the-art performance for Sequential Model Editing (SME). The code is available at `https://github.com/ZeroYuHuang/Transformer-Patcher`.

## 1 INTRODUCTION

Transformer-based models, particularly large Pretrained Language Models (PLMs) (Devlin et al., 2019; Brown et al., 2020) have become the backbone model of modern Natural Language Processing (NLP) and have enabled promising results in various downstream tasks (Lv et al., 2019; Budzianowski & Vulic, 2019; Ramnath et al., 2020). However, PLMs still produce undesirable outputs occasionally (Zhao et al., 2019; Basta et al., 2021). The cost of such mistakes is non-negligible. For example, a mistaken automatic translation result could get a person arrested (Hern, 2018). One of the most usual expedients was using a manual cache (e.g., lookup table) to overrule these problematic predictions (Sinitsin et al., 2020). Though convenient and straightforward, it lacks robustness and generality because it could be disabled by the slightest change in the input, such as paraphrasing in natural language. On the other hand, one can also re-train the model on the original dataset supplemented with problematic examples. While superior in performance, it is computationally and temporally expensive to re-train large PLMs with billions or even trillions of parameters.

Previous research formalized such problems as Model Editing (ME) and proposed various methods to intervene model's behavior on a specific example while preventing the model from forgetting other examples. Some straightly finetune the model on the example and used a constraint loss to maintain the model's overall performance (Zhu et al., 2020; Sotoudeh & Thakur, 2021). Some edit

the model through a HyperNetwork, which regards the model and the false predicted example as inputs and produced a weight update for the model's parameters (Cao et al., 2021; Sinitsin et al., 2020; Mitchell et al., 2022a). Despite their impressive progress, they mostly focus on one-step editing (fixing one mistake), which is not applicable to practical situations. Because models deployed for real-world applications are expected to face different errors ceaselessly. And the same error may pop up repeatedly and bother different users. In addition, as illustrated in Figure 1, once a wrong answer appears in an online question-answering (QA) model, leaving it unfixed and waiting for future corrections could mislead more people. Therefore, an ideal model editor should provide *continuous* and *promptly* fixing of newly emerged mistakes in an effective and efficient manner.

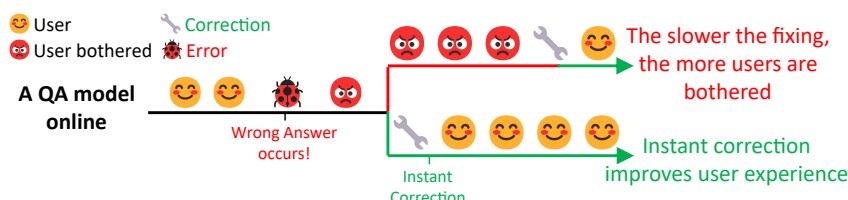

Figure 1: Once an error occurs in a QA model online, it could bother many users contacting the model if not fixed in time. Instant correction is a superior choice to improve the user experience, motivating us to propose a Sequential Model Editing problem.

Thus we extend the ME task into the sequential setting and formalize it as **Sequential Model Editing** (SME) task, which requires a model editor to fix a series of mistakes as soon as they appear. The desiderata of a qualified sequential model editor are three properties (Section 3). For each editing, the post-edit model should be of 1) **Reliability**: make the desirable output given the input; 2) **Generality**: generalize over other equivalent inputs; 3) **Locality**: retain its accuracy over irrelevant inputs. We then propose a standard SME experiment pipeline that is compatible with different tasks and five evaluation metrics to evaluate the three properties. Experiments show that most existing model editors could fail to generalize to the sequential editing scenario. Fine-tuning-based methods are vulnerable to forgetting previous edits. HyperNetwork-based editors are strongly coupled with the initial model that they are trained with, thus failing to edit the model after several steps (Section 5).

To handle SME, we introduce Transformer-Patcher. Unlike previous methods, Transformer-Patcher retains all original parameters to prevent harming the model's overall performance. It only adds a handful of trainable neurons (patches) to the last Feed-Forward Network (FFN) layer to revise the model's behavior on the problematic input and achieve a low editing cost. Furthermore, we train the patch to only respond to specific inputs with the proposed activation loss and memory loss. Experimental results on fact-checking (classification) and question answering (auto-regressive generation) indicated that Transformer-Patcher could rectify a series of mistakes (up to thousands) while almost perfectly retaining the model's overall performance.

The main contributions of this work are twofold: 1) We formally propose a sequential model editing task, as well as its standard experiment pipeline and evaluation metrics. 2) We introduce Transformer-Patcher, a simple yet effective model editor to revise transformer-based PLMs, achieving state-of-the-art SME performance.

## 2 RELATED WORKS

**Feed-forward Network**  Both the Transformer encoder and decoder contain the Feed-Forward Network (FFN). Recent works (Geva et al., 2021; Dai et al., 2022) analogously observed that FFN operates as key-value neural memories (Sukhbaatar et al., 2015). They regarded the input of FFN as a query, the first layer as keys, and the second as values. Thus the intermediate hidden dimension of FFN can be interpreted as the number of memories in the layer, and the intermediate hidden state is a vector containing activation values for each memory. Therefore, the final output of FFN can be viewed as the weighted sum of values activated.

**Model editors**  Existing model editors are mainly separated into two types: fine-tuning-based and HyperNetwork-based. Fine-tuning-based editors usually straightly tune the model with an extra loss to eschew over-fitting to edit examples. For instance, Zhu et al. (2020) proposed an extra loss

to reduce the distance between pre-edit and post-edit parameters. Mitchell et al. (2022a); Meng et al. (2022) equipped fine-tuning with KL-divergence to restrict the post-edit model's output space. For another, HyperNetwork-based editors require additional training phrases. Sinitsin et al. (2020) proposed a Meta Learning-based (Finn et al., 2017) approach named Editable Training to learn editable parameters for model modification. Cao et al. (2021) proposed KnowledgeEditor (KE) trained with constrained optimization to produce weight updates. Mitchell et al. (2022a) proposed MEND that learns to transform the gradient obtained by standard fine-tuning to edit large language models (Raffel et al., 2020). In addition, some works only focus on specific tasks, such as masked language modeling (Dai et al., 2022) and autoregressive language modeling (Meng et al., 2022; Geva et al., 2022). They require special input other than edit examples to conduct model editing.

**Continual Learning** The proposed SME task could be regarded as an emergent variant of Continual Learning (CL) (Mundt et al., 2020). And dynamically expandable networks are employed for CL as well (Rusu et al., 2016; Li & Hoiem, 2018). But there are some differences in the setting. In CL, usually, the model is continually trained using different datasets and tasks. But SME deals with only one example at once and all examples are from the same task. The difference in setting renders SME an unexplored area with new challenges that may not be properly addressed by general CL methods. For example, KL divergence loss and L2 normalization are usual methods to address the catastrophic forgetting in CL (De Lange et al., 2022), but previous works (Cao et al., 2021; Mitchell et al., 2022a) and our experiments show that they can hardly maintain models accuracy on irrelevant inputs in ME task. And methods that add task-specific parameters for CL usually need extra training (Yoon et al., 2018; Wortsman et al., 2020; de Masson d'Autume et al., 2019), thus falling short of SME's application efficiency requirement.

## 3 SEQUENTIAL MODEL EDITING PROBLEM

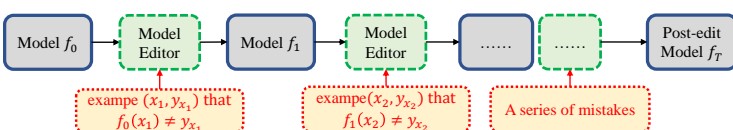

Figure 2: The process of sequential model editing task. Given the $t$-th mistake $(x_t, y_{x_t})$, the editor takes the model $f_{t-1}$ and $(x_t, y_{x_t})$ as input, and outputs the revised model $f_t$

.

Following Mitchell et al. (2022a), a model $f \in \mathbb{F}$ can be defined as a function $f : \mathbb{X} \mapsto \mathbb{Y}$ that maps an input $x$ to its prediction $f(x)$. Then, given a model $f$ and an edit example pair $(x_e, y_{x_e})$ that $f(x_e) \neq y_{x_e}$, a model editor ME is to output a post-edit model $f'$.

$$
\begin{aligned}
\text{ME}: \quad & \mathbb{F} \times \mathbb{X} \times \mathbb{Y} && \mapsto && \mathbb{F} \\
& (f, x_e, y_{x_e}) && \rightarrow && f' = \text{ME}(f, x_e, y_{x_e})
\end{aligned}
$$

Given a data stream $\{(x_1, y_{x_1}), \cdots, (x_s, y_{x_s})\}$ and an initial model $f_0$, a model editor ME needs to conduct edits successively when the model makes undesirable output, as shown in Figure 2.

$$
f_t = \begin{cases}
f_0 & \text{if } t = 0, \\
f_{t-1} & \text{elif } f_{t-1}(x_t) = y_{x_t}, \\
\text{ME}(f_{t-1}, x_t, y_{x_t}) & \text{else.}
\end{cases}
\tag{1}
$$

And after every edit in SME the post-edit model $f'$ should satisfy the following three properties:

**Property 1** *Reliability: the post-edit model should output the desired prediction:*

$$
f'(x_e) = y_{x_e}
\tag{2}
$$

**Property 2** *Generality: given an edit example $x_e$, $\mathbb{E}_{x_e} = \{x_j | y_{x_j} = y_{x_e}\}$ is defined as the set of its equivalent inputs (e.g. rephrased sentences). Then the post-edit model $f'$ should satisfy:*

$$
\forall x_j \in \mathbb{E}_{x_e}, f'(x_j) = y_{x_e}
\tag{3}
$$

**Property 3** *Locality: the edit should be implemented locally and precisely, which means the post-edit model should remain accurate on the irrelevant examples set $\mathbb{I}_{x_e} = \mathbb{X} \backslash \mathbb{E}_{x_e}$:*

$$\forall x_j \in I_{x_e}, f'(x_j) = y_{x_j} \tag{4}$$

*In particular, an edit should not disrupt the results of past edits in SME setting, which means:*

$$f_t(x_k) = y_{x_k}, \text{ for } k \text{ where } f_{k-1}(x_k) \neq y_{x_k} \tag{5}$$

## 4 TRANSFORMER-PATCHER

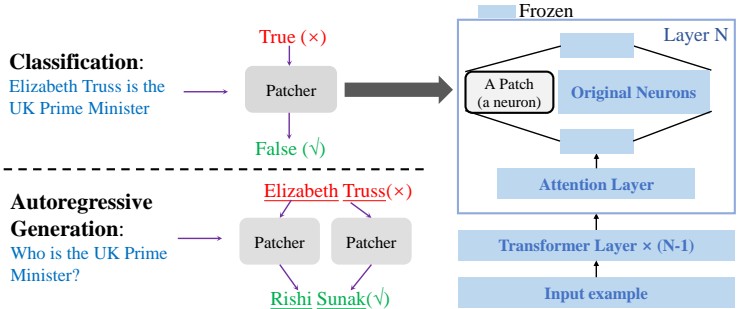

Figure 3: Transformer-patcher enables efficient correction for classification and generation tasks, it rectifies the model's behavior by adding and training several extra neurons in the last FFN layer.

First, we call one misclassification or one wrongly generated token one *mistake* in the rest of the paper. Aiming at the SME task for transformer-based models, we propose **Transformer-Patcher** shown in Figure 3. It freezes all original parameters and adds one neuron (patch) to the last FFN layer for one mistake. And we train the patch to take effect only when encountering its corresponding mistake. For classification, we add only one patch to rectify the model. For auto-regressive generation, we count how many tokens are wrongly generated under the teacher-forcing setting and add one patch for each of them. This section describes how to add and train one patch. Multiple patch editing follows exactly the same principle and is formally described in Appendix A.

### 4.1 WHAT IS A PATCH?

As mentioned in Section 2, FFN operates as key-value neuron memories. Its forward computation is a process that retrieves values from matrix $V$ by matching keys in matrix $K$ and the input query $q$. For a standard FFN, given a query $q \in \mathbb{R}^d$, its output $FFN(q)$ is:

$$a = \text{Act}(q \cdot K + b_k) \tag{6}$$
$$FFN(q) = a \cdot V + b_v \tag{7}$$

where $\text{Act}(\cdot)$ is a non-linear activation function (e.g., Relu or Gelu), $a$ is the vector of activation values, $b_k$ and $b_v$ are two bias vectors. A patch is an extra neuron (an extra key-value pair) added to the last FFN layer. After patching, the new output $FFN_p(q)$ is:

$$[a \quad a_p] = \text{Act}(q \cdot [K \quad k_p] + [b_k \quad b_p]) \tag{8}$$

$$FFN_p(q) = [a \quad a_p] \cdot \begin{bmatrix} V \\ v_p \end{bmatrix} + b_v \tag{9}$$

where $k_p \in \mathbb{R}^d$ is the patch key, $v_p \in \mathbb{R}^d$ is the patch value, $b_p$ is a scalar named patch bias, $a_p = \text{Act}(q \cdot k_p + b_p)$ represents the activation value of the patch. With the substitution of equations 6 and 7, equation 9 can be reformulated as:

$$FFN_p(q) = FFN(q) + a_p \cdot v_p \tag{10}$$

### 4.2 TRAINING A PATCH FOR EDITING

An ideal edit requires **reliability**, **generality**, and **locality** proposed in Section 3. For **reliability**, a patch needs to be activated according to equation 10. Let $q_e$ represent the input query of the mistake,

the patch key $\boldsymbol{k}_p$ and patch bias $b_p$ should satisfy:

$$a_p = \text{Act}(\boldsymbol{q}_e \cdot \boldsymbol{k}_p + b_p) \neq 0 \tag{11}$$

When Act is ReLU or GeLU, the above condition can be approximated as follows:

$$\boldsymbol{q}_e \cdot \boldsymbol{k}_p + b_p > 0 \tag{12}$$

To meet the constraint 12, we propose a activation loss $l_a$ to maximize the activation value:

$$l_a = \exp(-\boldsymbol{q}_e \cdot \boldsymbol{k}_p - b_p)) \tag{13}$$

Once a patch is activated, according to equation 10, it adds a bias term $a_p \cdot \boldsymbol{v_p}$ to the output of the last layer. Because we are editing the last layer of the model, the output of the model can be adjusted to any result without worrying that other components of the model would cancel the editing effect. To obtain the target output, we leverage the task's original loss function and rename it as edit loss $l_e$. Formally, for an edit example $(x_e, y_e)$, the patched model's output is $p_e$, $l_e$ is defined as:

$$l_e = L(y_e, p_e) \tag{14}$$

where $L(\cdot)$ is a function of label $y_e$ and model output $p_e$ and depends on the specific task.

For **locality**, the model's behavior should not be shifted on irrelevant examples, thus the patch should not be activated by any irrelevant examples. When using ReLU or GeLU, it can be approximated as that all queries from irrelevant examples $\boldsymbol{q}_i$ should have a patch activation value less than or equal to a threshold $\beta$, i.e., the maximum of them is less than or equal to $\beta$:

$$\forall i \in \mathbb{I}_{x_e}, \boldsymbol{q}_i \cdot \boldsymbol{k}_p + b_p \leq \beta \rightarrow \max_i(\boldsymbol{q}_i \cdot \boldsymbol{k}_p + b_p) \leq \beta \tag{15}$$

Thus we propose the memory loss $l_m$ to enforce the constraint 15. To imitate the distribution of queries from irrelevant examples, we randomly retain some queries from previously seen examples as memories. Each query is a $d$-dimensional vector and we can stack them as a matrix $\boldsymbol{M} \in \mathbb{R}^{d_m \times d}$, where $d_m$ is the number of queries saved. Our proposed memory loss $l_m$ is the sum of two terms. The first term $l_{m1}$ is introduced to make the patch inactivated to all queries in $\boldsymbol{M}$:

$$l_{m1} = S(\boldsymbol{M} \cdot \boldsymbol{k}_p + b_p - \beta; k) \tag{16}$$

where $S(\cdot; k)$ is a function that receives a vector $\boldsymbol{v}$ and outputs a scalar

$$S(\boldsymbol{v}; k) = \text{Avg}[\text{TopK}(\exp(\boldsymbol{v}); k)] \tag{17}$$

It first employs element-wise exponential function to $\boldsymbol{v}$ and then selects $k$ largest elements to compute their average as the output. Although constraint 15 is about the maximum, we employ TopK here for more efficient optimization. In case that $l_{m1}$ can not absolutely ensure the constraint 15, we propose $l_{m2}$ to distance the activation value of $\boldsymbol{q}_e$ and $\boldsymbol{q}_i$. That is, the activation value of the mistaken example is larger than that of the irrelevant examples by a certain margin $\gamma$.

$$l_{m2} = S((\boldsymbol{M} - \boldsymbol{q}_e) \cdot \boldsymbol{k}_p + b_p - \gamma; k) \tag{18}$$

To sum up, the loss $l_p$ for training a patch is defined as a weighted sum of the above losses:

$$l_p = l_e + al_a + ml_m = l_e + al_a + m(l_{m1} + l_{m2}) \tag{19}$$

where $a$, $m$ are hyper-parameters. $\beta$ is selected as -3 for GeLU and 0 for ReLu, since GeLU(-3)$\approx$0.004 is small enough and ReLU(0)=0. $\gamma$ is selected as 3 for GeLU and 0 for ReLU.

## 5 EXPERIMENTS

### 5.1 EXPERIMENTAL SETTINGS AND EVALUATION METRICS

We proposed an experimental pipeline for SME used for standard datasets with training set $\mathbb{D}_{train}$, validation set $\mathbb{D}_{val}$, and test set $\mathbb{D}_{test}$. There are two differences between our setting and the previous Model Editing setting. First, we employ multi-step editing rather than one-step. Second, previous works usually generate counterfactual edit examples (e.g., replacing the answer to a question with a random one), while we employ authentic examples where the model makes mistakes. We first split

the original $\mathbb{D}_{train}$ into an edit set $\mathbb{D}_{edit}$ and a new training set $\mathbb{D}'_{train}$. To evaluate generality, back-translation could be utilized to generate the equivalent set $\mathbb{E}_{x_e}$ for edit example $x_e \in \mathbb{D}_{edit}$ following previous works (Cao et al., 2021). To evaluate locality, a subset $\mathbb{D}_{tr}$ randomly sampled from $\mathbb{D}'_{train}$ is used to see how the post-edit model performs on its training data. Our SME pipeline starts with an initial model $f_0$ trained on $\mathbb{D}'_{train}$ and validated using $\mathbb{D}_{val}$, the model is sequentially edited while encountering mistakes in $\mathbb{D}_{edit}$. After the $t$th edit example $(x_e^t, y_e^t)$, we obtain a post-edit model $f_t$. Supposing that there are $T$ total edits and $I$ represents the indicator function, our proposed metrics are calculated as follows:

1) **Success Rate** (SR): to evaluate the reliability, we test if the post-edit model outputs the desired prediction. Thus, SR is:

$$SR = \frac{1}{T} \sum_{t=0}^{T} I(f_t(x_e^t) = y_e^t) \tag{20}$$

2) **Generalization Rate** (GR): to evaluate the generality, we test the post-edit model $f_t$ on the equivalent set $\mathbb{E}_{x_e^t} = \{x_{e,1}^t \cdots, x_{e,N_t}^t\}$ of the edit example $x_e^t$, thus GR is:

$$GR = \frac{1}{TN_t} \sum_{t=0}^{T} \sum_{i=1}^{N_t} I(f_t(x_{e,i}^t) = y_e^t) \tag{21}$$

3) **Edit Retain Rate** (ER): to evaluate locality and reliability, we evaluate how many past edits are retained by the final model $f_T$. In a real application, a reliable model editor should keep the fixed bugs from recurring again, thus SR alone cannot evaluate reliability, and we define ER by testing the final model on all its past edit examples $\mathbb{E}_{pe}$:

$$ER = \frac{1}{T} \sum_{t=0}^{T} I(f_T(x_e^t) = y_e^t)/T \tag{22}$$

4) **Training Retain Rate** (TrainR): to evaluate locality, we compare the performance of the final model of $f_T$ and the initial model $f_0$ on subsampled test $\mathbb{D}_{tr}$. Thus, the TrainR is defined as:

$$TrainR = \frac{\sum_{(x,y) \in D_{tr}} I(f_T(x) = y)}{\sum_{(x,y) \in D_{tr}} I(f_0(x) = y)} \tag{23}$$

5) **Test Retain Rate** (TestR): to evaluate locality, we see if the post-edit model still retains the generalization ability over unseen data. Then the TestR is defined as:

$$TestR = \frac{\sum_{(x,y) \in D_{test}} I(f_T(x) = y)}{\sum_{(x,y) \in D_{test}} I(f_0(x) = y)} \tag{24}$$

**Datasets and Baselines** Both classification and auto-regressive generation tasks are selected for evaluation. Following Cao et al. (2021) and Mitchell et al. (2022a), we employ Fact-Checking (FC) for classification and closed-book Question Answering (QA) for generation. For FC, we apply a BERT base model (Devlin et al., 2019) and the FEVER dataset (Thorne et al., 2018). For QA, we apply a BART base model (Lewis et al., 2020) and the Zero-Shot Relation Extraction (zsRE) dataset (Levy et al., 2017). We directly use the equivalent set released by Cao et al. (2021). We use the same data split as Cao et al. (2021). Both FC and QA are evaluated using accuracy. Our baselines include (1) **Fine-Tuning-based editors**: The **FT** directly fine-tunes the model on the edit example. Following Mitchell et al. (2022a), **FT+KL** is selected as a baseline. It fine-tunes the model with an extra KL divergence loss $l_{kl}$. Following Sinitsin et al. (2020) and Zhu et al. (2020), we report fine-tuning-based baselines by fine-tuning all parameters (**FT(all)** and **FT(all)+KL**) or the last layer (**FT(last)** and **FT(last)+KL**). (2) **Two HyperNetwork-based editors: KE** (Cao et al., 2021) and **MEND** (Mitchell et al., 2022a). (3) **SERA**: a variant of the latest SOTA memory-based model editor SERAC (Mitchell et al., 2022b). Other details of our baselines are reported in Appendix B.

**Experiment Details** Initial models for two tasks are obtained following the same training settings as Cao et al. (2021). For FC, the accuracy of the initial model attains 94.1% on $\mathbb{D}_{tr}$, 76.9% on $\mathbb{D}_{test}$. For QA, the accuracy of the initial model attains 56.6% on $\mathbb{D}_{tr}$, 23.1% on $\mathbb{D}_{test}$. To reduce the experimental uncertainty, we randomly split the edit set into $n = 20$ folders to run SME 20

Table 1: The Success Rate (SR), Generalization Rate (GR), Edit Retain Rate (ER), Training Retain Rate (TrainR), Test Retain Rate (TestR) of Transformer-Patcher (T-Patcher) and the baselines on FEVER and zsRE dataset. * denotes that the SR of the T-patcher on QA is 0.9987. † means the method requires extra training phases and training data.

| Editor | FEVER Fact-Checking BERT-base (110M) | | | | | zsRE Question-Answering BART-base (139M) | | | | |
| | SR | GR | ER | TrainR | TestR | SR | GR | ER | TrainR | TestR |
|---|---|---|---|---|---|---|---|---|---|---|
| FT(last) | **1.00** | 0.61 | 0.59 | 0.893 | 0.946 | **1.00** | 0.58 | 0.30 | 0.914 | 0.924 |
| FT(all) | **1.00** | 0.74 | 0.83 | 0.968 | 0.994 | **1.00** | 0.68 | 0.43 | 0.865 | 0.910 |
| FT(last)+KL | **1.00** | 0.53 | 0.45 | 0.968 | 0.998 | **1.00** | 0.57 | 0.28 | 0.923 | 0.933 |
| FT(all)+KL | **1.00** | 0.71 | 0.49 | 0.998 | **1.011** | **1.00** | 0.68 | 0.39 | 0.889 | 0.925 |
| MEND† | 0.04 | 0.03 | 0.06 | 0.349 | 0.652 | 0.41 | 0.37 | 0.00 | 0.000 | 0.000 |
| KE† | 0.14 | 0.12 | 0.28 | 0.486 | 0.650 | 0.09 | 0.08 | 0.00 | 0.000 | 0.000 |
| SERA† | **1.00** | **0.89** | **1.00** | 0.904 | 0.916 | **1.00** | **0.90** | 0.98 | 0.906 | 0.901 |
| T-Patcher | **1.00** | 0.82 | **1.00** | **0.999** | 1.000 | 1.00* | 0.82 | **0.99** | **0.997** | **0.996** |

Table 2: The experimental results when utilizing all data in $D_{edit}$ as a single run of SME on QA task. The results of the FC task are presented in Table 7 in Appendix C. E represents how many edits have been conducted. N represents how many mistakes have been made by the initial model $f_0$ on the entire edit set $D_{edit}$.

| Editor | SR | GR | ER | TrainR | TestR | E | N |
|---|---|---|---|---|---|---|---|
| **FT(all)+KL** | 1.00 | 0.69 | 0.14 | 0.936 | 0.974 | 2821 | 2766 |
| **SERA** | 1.00 | 0.90 | 0.97 | 0.728 | 0.694 | 3558 | 2766 |
| **T-Patcher** | 0.99 | 0.81 | 0.97 | 0.912 | 0.948 | 2308 | 2766 |

times and report the averaged performance as the final result. The initial model $f_0$ makes about 63 mistakes in an FC folder and about 139 in a QA folder on average. For methods requiring memories (fine-tuning with KL and ours), 40,000 memory examples are sampled from $D'_{train} \setminus D_{tr}$ are employed for both tasks and are updated as editing proceed. The hyperparameters $a$ and $m$ are selected as 1 and 10 respectively for both tasks to make the extra losses and the original task loss in the same order of magnitude. Other details can be found in Appendix B.

## 5.2 EXPERIMENTAL RESULTS

**Main results** The experiment results are shown in Table 1. Our method achieves strong performance in all five metrics across two tasks. It could make a series of model corrections (SR≈1) while nearly retaining every past edit (ER≈1) and almost perfectly keeping the model's overall performance (TrainR≈1, TestR≈1). The fine-tuning-based editors could partly preserve the model's behavior and achieve high SR, but it is vulnerable to forgetting previous edits (low ER). Two HyperNetwork-based editors fail in the SME setting. They have trouble retaining models' overall performance (low ER, TrainR, TestR) and conducting a series of edits (low SR and GR). SERA achieves the highest GR, while can only partially preserve the model's overall performance (TestR, TrainR≈0.9) compared to T-Patcher. Apart from being effective, our method is efficient enough as well. Using a V100, one edit costs only 7.1s for FC and 18.9s for QA. We could further improve the efficiency to 4.7s and 12.4s by decreasing the number of memory examples to 10,000.

**Scale up to thousands of edits** Table 1 shows that Transformer-Patcher achieves good performance for about 60 edits on FC and 140 edits on QA, thus we wonder if it could handle more edits. So we utilize all data in $D_{edit}$ as a single data stream to run SME. As shown in Table 2, Transformer-Patcher could effectively correct up to thousands of mistakes and retain the model's overall performance simultaneously compared with the other two strong baselines. It's interesting to notice that the number of edits E of Transformer-Patcher is less than the number of actual mistakes N made by the initial model. In other words, our method can fix some potential mistakes in the initial model before the error actually happens. On the contrary, the fine-tuning-based method fixes more mistakes than the original model, which means it created more errors during the editing

process. It seems contradictory that our method attains fewer E and lower TestR, this may due to the distribution shift between $\mathbb{D}_{edit}$ and $\mathbb{D}_{test}$. See more explanation in Appendix C. Furthermore, the post-edit model only gets **1.4%** larger for FC and **4.5%** larger for QA. We believe this cost is acceptable for automatically correcting the model's mistakes from time to time during deployment. In practice, we suggest using the transformer-patcher to provide a timely response for each mistake online, and after accumulating certain quantities of mistakes, we could fine-tune the original model on all accumulated mistakes, so that the patches can be removed. In this way, we could achieve a good balance between model size and editing effectiveness.

## 5.3 ANALYSES

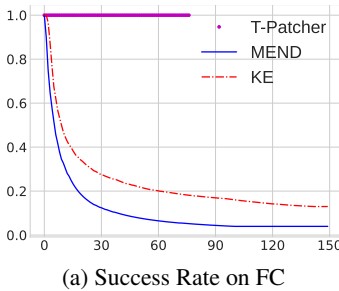

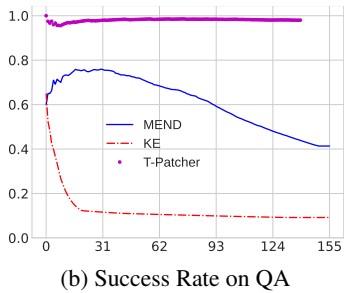

(a) Success Rate on FC

(b) Success Rate on QA

Figure 4: Variation of success rate (SR) with the number of edits. Different methods have different edit times, we plot until they converge.

**The collapse of MEND and KE** We discuss here why MEND and KE fail in the SME. Figure 4 presents how SR varies with the number of edits on both FC and QA. Figure 4 shows that MEND and KE are effective in the first few steps, but shortly after they are no longer able to produce valid edits. However, in their original paper (Cao et al., 2021; Mitchell et al., 2022a), they both reported that they achieved high SR when dealing with one-step editing. We find this phenomenon reasonable since both HyperNetwork-based editors are trained with the initial model $f_0$ and thus strongly coupled with the original parameters. As the editing proceeds, the model becomes more different from the initial one, resulting in their failure. We tried to retrain HyperNets after every edit using the post-edit model, but the cost for re-training is unacceptable as it costs hours to train a HyperNet model editor.

Table 3: The ablation results for two alternatives of memory loss.

| Patch | FEVER Fact-Checking | | | | | zsRE Question-Answering | | | | |
|---|---|---|---|---|---|---|---|---|---|---|
| | SR | GR | ER | TrainR | TestR | SR | GR | ER | TrainR | TestR |
| w/o $l_m$ | 0.99 | **0.94** | 0.61 | 0.737 | 0.844 | 0.99 | **0.94** | 0.21 | 0.069 | 0.154 |
| KL | **1.00** | 0.76 | 0.99 | 0.996 | 0.998 | 0.94 | 0.69 | 0.49 | 0.481 | 0.710 |
| w/o $l_{m_2}$ | 0.95 | 0.82 | 0.95 | 0.994 | 0.992 | 0.95 | 0.82 | 0.94 | 0.991 | 0.984 |
| T-Patcher | **1.00** | 0.82 | **1.00** | **0.999** | 1.000 | **1.00** | 0.82 | **0.99** | **0.997** | **0.996** |

**Memory loss** To validate the effectiveness of our proposed memory loss, we apply several alternative patches: (1) T-Patcher w/o $l_m$, (2) KL Patch, where $l_m$ is replaced with the KL divergence loss, (3) T-Patcher w/o $l_{m_2}$. The ablation results in Table 3 show that memory loss is critical. Simply adding patches without memory loss hurts the model's overall performance severely. The KL divergence partially alleviates this problem (higher TrainR, TestR, and ER) but is still unsatisfying on the more complex QA task, which is similar to the Fintuning with KL results in Table 1. By comparing w/o $l_{m_2}$ and T-Patcher, we observe that the main contribution of our proposed memory loss comes from $l_{m_1}$, while adding $l_{m_2}$ still improves the method's performance. Furthermore, to investigate whether our added patches do solely respond to the specific error we visualize the activation values of different patches on their corresponding mistakes in Figure 5 for the QA task. The X-axis represents the mistake (8.2 represents the second mistake of the 8th edit example) and the Y-axis represents the patch. Figure 5a shows that the patch can be activated by multiple irrelevant queries without the constraint of memory loss, leading to low ER, TrainR, and TestR. Figure 5b is a lot darker, indicating that the KL loss tends to deactivate patches to bridge the distribution gap

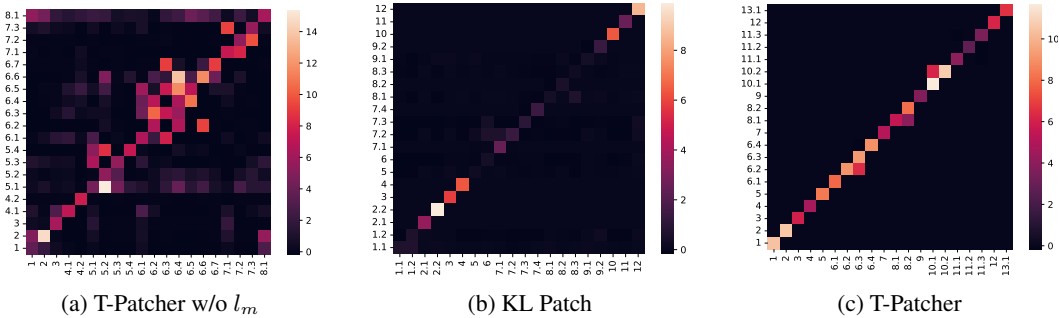

| (a) T-Patcher w/o $l_m$ | (b) KL Patch | (c) T-Patcher |

Figure 5: The activation values of three different patches on their corresponding mistakes.

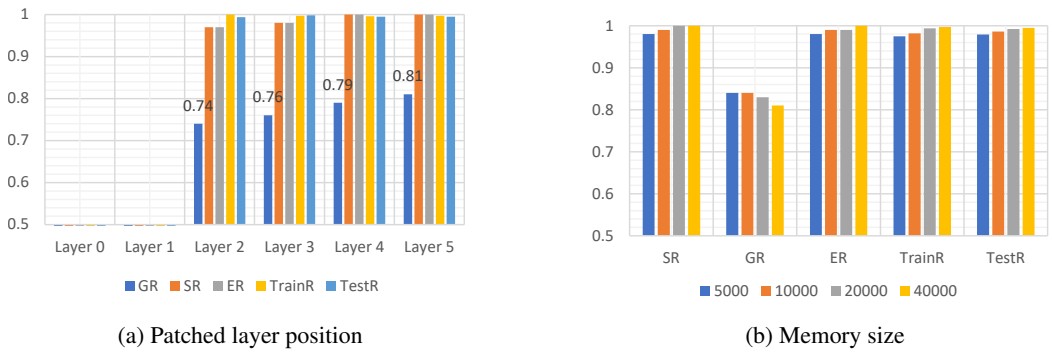

| (a) Patched layer position | (b) Memory size |

Figure 6: The ablation studies about patched layer position and the memory size .

before patching and after patching. And figure 5c presents a clear diagonal line, which means each patch takes charge of its corresponding mistake. Further analysis of the activation value of different patches is presented in Appendix C.

**Patched layer position**   To validate the benefits of patching the last layer, we focus on the QA task and patch each decoder layer separately. The ablation results are illustrated in Figure 6a. First, patching the bottom layer (layer 0 and 1) can not make effect edits. This may be because patching the bottom layer severely influences every token in the input sequence, making the patch's optimization more difficult. While the patches added to the last layer only influence correspondent mistaken tokens, Then, compared to the other metrics, what the patching position influenced most is GR, which increases from 0.74 of layer 2 to 0.81 of layer 5, proving that patching the top layers may improve the **generality**. This phenomenon is aligned with previous studies  (Jawahar et al., 2019) which found that high-level semantic features are encoded at the top layers and superficial information is encoded in lower layers. Besides, patching the last layer could ameliorate the editing efficiency as well. Because computation results of previous layers could be cached and reused while editing.

**Memory size**   In order to verify the robustness of our method, we conduct experiments using different memory sizes (from 5,000 to 40,000) on the QA task. As is shown in Figure 6b, our method is not very sensitive to the size of the memory set. Reducing memory examples only causes slight drops in SR, ER, TrainR, and TestR, and a slight increase in GR.

## 6   CONCLUSION

In this work, we proposed the Sequential Model Editing task, as well as its experiment pipeline and evaluation metrics. We then introduce Transformer-Patcher, a practical method for sequentially editing transformer-based language models. Experiments on both classification and autoregressive generation tasks demonstrate its ability to edit the model up to a thousand times continuously. This method could have a positive social impact by fixing serious mistakes in large PLMs, including generating biased predictions and hate speech, benefiting a broad spectrum of audiences.

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

# A  MULTIPLE NEURON PATCHING

In auto-regressive generation tasks, the model may make multiple mistakes in one example. Since FFN is a position-wise network, every mistake in the output can be ascribed to one query to the last FFN layer. Therefore, for an example where the model makes $n$ mistakes, each mistake can be ascribed to a query $q_e^i$ to the last FFN layer, and we add $n$ patches to handle each of them. Specifically, given an input query $q$, the new output $FFN_p(q)$ of a FFN with $n$ patches is:

$$[a \quad a_p] = \mathrm{Act}(q \cdot [K \quad K_p] + [b_k \quad b_p]) \tag{25}$$

$$FFN_p(q) = [a \quad a_p] \cdot \begin{bmatrix} V \\ V_p \end{bmatrix} + b_v \tag{26}$$

where $K_p \in \mathbb{R}^{d \times n}$ is the patch key, $v_p \in \mathbb{R}^{n \times d}$ is the patch value, $b_p \in \mathbb{R}^n$ is the patch bias, $a_p = \mathrm{Act}(q \cdot k_p + b_p)$ is a vector containing activation values of patches. With the substitution of equations 6 and 7, equation 9 can be reformulated as:

$$FFN_p(q) = \begin{cases} FFN(q) & \text{if } a_p = \vec{0} \\ FFN(q) + a_p \cdot v_p & \text{else} \end{cases} \tag{27}$$

During calculating the activation loss for multiple patches, we just constraint the patch $k_p^i$ to be activated by its corresponding query $q_e^i$, let $q_e \in \mathbb{R}^{n \times d}$ represent the matrix containing $n$ corresponding queries, then we can obtain $\mathcal{A} \in \mathbb{R}^n$ which is defined as a vector containing activation values of each patch on its corresponding query:

$$\mathcal{A}_i = q_e^i \cdot k_p^i + b_p^i \tag{28}$$

It can also be formulated as follows:

$$\mathcal{A} = \mathrm{diag}(q_e \cdot k_p) + b_p \tag{29}$$

where $\mathrm{diag}$ is a function to select the diagonal elements from a matrix. Then the activation loss for $n$ patches can be calculated as follows:

$$l_a = S(-\mathcal{A}; k_a) \tag{30}$$

where $S$ is the function defined in Equation 17, $k_a$ is a hyper-parameter.

Memory loss $l_m$ for multiple patches remains the sum of two terms $l_{m1}$ and $l_{m2}$, where $l_{m1}$ is identical as Equation 16. As for $l_{m2}$, we restrict that for $i$-th patch $k_p^i$, all its activation value to a query in $M$ should be smaller than that to its corresponding query $q_e^i$, thus $l_{m2}$ becomes:

$$l_{m2} = S(M \cdot k_p + b_p - \mathcal{A} - \gamma; k) \tag{31}$$

For initialization, every patch $k_p^i$ is initialized as its normalized related query $\frac{q_e^i}{|q_e^i|^2}$ so that the initial activation value is 1.

# B  EXPERIMENTAL DETAILS

**Data splits**   We utilize the same data split of training and testing following Cao et al. (2021). For closed-book fact-checking, the binary FEVER dataset originally has 104,966 training instances and 10,444 validation instances. In order to adapt it to the SME task, we keep the original validation set intact and employ it as $\mathbb{D}_{test}$, and split the original training data into three subsets: a new training set $\mathbb{D}'_{train}$, a new validation set $\mathbb{D}_{val}$ and an edit set $\mathbb{D}_{edit}$ in the ratio of $0.8 : 0.1 : 0.1$. As a result, we get 10,496 instances for the edit set. Since the Bert-based classifier attains 88.3% on the edit set, the ideal edit sequence length is 10496*88.3%/20=63 on average.

For closed-book question answering, we employ the zsRE dataset released by Cao et al. (2021), which originally has 244,173 examples for training and 27,644 examples for validation. We first filter out examples with only one answer and then employ the same data split process as FEVER in the ratio of $0.9 : 0.075 : 0.025$. Finally, we get 5,317 edit data and 15,982 for validation, and 24,051 for testing. Since the Bart-based model attains 47.9% on the edit set, the ideal edit sequence length is 5317*47.9%/20=139 on average. For both datasets, we randomly sampled a subset from $\mathbb{D}'_{train}$ with the size of 10,000 as $\mathbb{D}_{tr}$, and the edit set $\mathbb{D}_{edit}$ is split into $n = 20$ folders to run SME $n = 20$ times independently. For the model editor requiring memories (fine-tuning with KL and Transformer-Patcher), we randomly sampled a subset from $\mathbb{D}'_{train} \setminus \mathbb{D}_{tr}$ with the size of 40000 and update it as the editing proceeds.

**Initial models training**  Initial models are trained following Cao et al. (2021). For the Fact-Checking task, we fine-tune a BERT base model with an additional linear layer that maps the hidden state of the BOS (beginning of a sentence) token to the probability of the positive label. We maximize the model likelihood and the final model attains an accuracy of 76.9% on $\mathbb{D}_{test}$, 94.1% on $\mathbb{D}_{tr}$ and 88.3% on $\mathbb{D}_{edit}$. For the QA task, we fine-tune a BART base model by maximizing the model likelihood regularized with dropout and label smoothing. The final model attains an accuracy (exact match between model prediction and ground truth) of 23.1% on $\mathbb{D}_{test}$ , 56.6% on $\mathbb{D}_{tr}$ and 47.9% on $\mathbb{D}_{edit}$. And these results are comparable with results that the model trained and released by Cao et al. (2021) has achieved.

**Transformer-Patcher training details**  For FC, we add one patch for every edit example. For QA, we employ the teacher forcing setting and count how many target tokens are not assigned to the highest likelihood as the mistake number. For one edit example, we add up to 5 patches.FC and QA task share almost the same hyper-parameters. We repeat one edit example 8 times and feed them to Transformer-Patcher as a batch for training. The initial learning rate is set as 0.01. Adam optimizer (Kingma & Ba, 2015) is applied for both tasks. Every patch is initialized with the normalized corresponding query $\frac{q_e}{|q_e|^2}$. Such a method makes each patch activated with an initial activate value 1. The patch value $v_p \in \mathbb{R}^{n \times d}$ is parameterized as element-wise production of two matrices: $v_p' \in \mathbb{R}^{n \times d}$ and $n_p \in \mathbb{R}^{n \times d}$, $v_p'$ is initialized with the random number between 0 and 1, and elements in $n_p$ is initialized with an integer 5 to make the patch value dominant over existing values $V$.The parameter $k_a$ mentioned in equation 30 is set as 5, and parameter $k$ for memory loss is set as 1000. All hyper-parameters are chosen by running a few examples on the validation set.

**Baseline implementation details**  For KE, we directly utilize the released trained HyperNetwork for conducting SME experiments (Cao et al., 2021).

For MEND, there is no HyperNetwork released and we re-implement the released code with hyper-parameters set as Mitchell et al. (2022a). We employ fine-tuning-based methods following Mitchell et al. (2022a) and Cao et al. (2021).

For all fine-tuning-based baselines, we set the learning rate as 1e-5 and utilize Adam's optimizer to fine-tune the model until the mistaken example is corrected. For the computation of KL loss for fine-tuning +KL-constraints baselines, we randomly sample a batch of examples in a memory set with the size of 512.

For SERAC (Mitchell et al., 2022b), we implement one variant of it: SERA. The SERAC maintains a cache of all edit examples. Given an input, it first employs a scope classifier to estimate if the input is relevant to (falls in the scope of) any cached edit examples. If so, it then employs a counterfactual model (needs to have the identical output space as the original model) to produce the output relying on the most relevant cached example. Otherwise, it returns the output of the original model. In our proposed SME experiment setting, the in-scope examples have the same label as the edit example, thus the function of the counterfactual model is to reproduce the answer of the relevant example. During the implementation of QA, we choose the Bart-base as the counterfactual model, but we find is not trivial for the Bart model to reproduce the answer (the original paper use T5 for generation tasks), thus it is more practical to directly return the label of the cached edit example. We refer to this direct-return method as SERA and include it as our baseline for both Fact-Checking and Question-Answering tasks. All other implementation details about SERA are the same as the original paper (Mitchell et al., 2022b).

**Environment details**  For all methods, we run SME experiment $n$=20 times on $n$ different edit folders simultaneously using 8 NVIDIA Tesla V100 GPUs. And it cost around 1 hour for running Trnasformer-Patcher on FEVER and around 3 hours on zsRE.

## C  EXTRA EXPERIMENT RESULTS

**Variation of locality with the number of edits**  The metric ER, TestR, and TrainR reflect the locality of the final model, but how models behave in the middle is still unclear to us. Thus we choose KE, MEND, FT(all)+KL, and Transformer-Patcher and investigate how their locality varies with the number of edits on the QA task. The results are shown in Figure 7. As editing continues,

Table 4: Mean and deviation of absolute patches activation values on three different kinds of examples

| Patch | FEVER Fact-Checking | | | zsRE Question-Answering | | |
|---|---|---|---|---|---|---|
| | Edit | Past-edit | Random | Edit | Past-edit | Random |
| w/o $l_m$ | 34.3±9.3 | 15.7±8.1 | 0.5±3.0 | 11.32±7.3 | 1.23±1.64 | 0.14±0.3 |
| KL | 9.15±2.7 | 0.01±0.16 | 0.05±0.2 | 1.12±1.87 | 0.03±0.06 | 0.12±0.1 |
| T-Patcher | 10.25±2.3 | 0.00±0.0 | 0.05±0.1 | 6.78±2.58 | 0.00±0.00 | 0.10±0.1 |

Table 5: The standard deviation of Edit Retain Rate (ER), Training Retain Rate (TrainR), Test Retain Rate (TestR) of Transformer-Patcher (T-Patcher) and fine-tuning based baselines on FEVER and zsRE dataset.

| Editor | FEVER Fact-Checking BERT-base (110M) | | | zsRE Question-Answering BART-base (139M) | | |
|---|---|---|---|---|---|---|
| | ER | TrainR | TestR | ER | TrainR | TestR |
| FT(last) | 0.05589 | 0.06242 | 0.03322 | 0.03981 | 0.00920 | 0.01860 |
| FT(all) | 0.07008 | 0.03368 | 0.02178 | 0.05168 | 0.02322 | 0.01781 |
| FT(last)+KL | 0.05929 | 0.02516 | 0.01635 | 0.03173 | 0.01293 | 0.01697 |
| FT(all)+KL | 0.06248 | 0.00677 | 0.01116 | 0.06433 | 0.01659 | 0.01953 |
| **T-Patcher** | 0.00000 | 0.00045 | 0.00048 | 0.00916 | 0.00101 | 0.00115 |
| w/o $l_m$ | 0.10332 | 0.21872 | 0.14569 | 0.23259 | 0.05063 | 0.15795 |
| KL | 0.00078 | 0.00536 | 0.00248 | 0.07124 | 0.08237 | 0.02469 |

more and more damage has been done to the model by other baselines, except Transformer-Patcher.

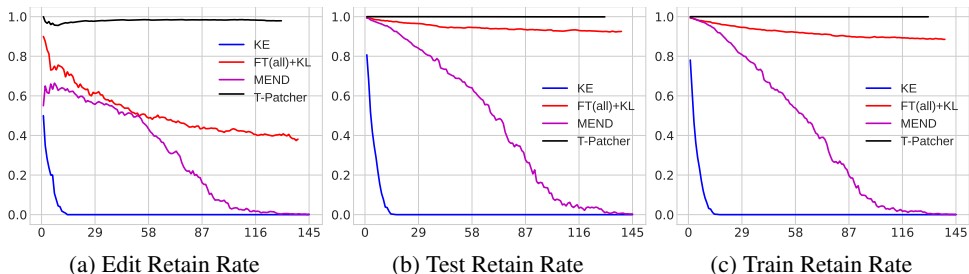

(a) Edit Retain Rate      (b) Test Retain Rate      (c) Train Retain Rate

Figure 7: Variation of ER, TestR, and TrainR with the number of edits on QA task.

**Standard deviation of experiment results** Since some values in Table 1 and Table 3 are very close, we report the standard deviation in Table 5. Note that the SR and the GR are calculated using all different folders at the same time, the standard deviation is therefore 0. According to Table 5, Transformer-Patcher achieves the smallest deviation on ER, TrainR, and TestR.

**Statistics of activation values of different patches** In order to study the activation situation of patches on different examples. we present the mean and deviation of absolute patches activation values on three different mistakes: 1) Edit: the mistake for which the patch is added; 2) Past-edit: mistakes from previous edit examples; 3) Random: mistake of examples randomly sampled from $D_{test}$. As BERT and BART utilize GeLU, both positive and negative activation values could activate the patch. We employ absolute value to measure to what extent the patch is activated. The results are shown in Table 4. First, the T-Patcher w/o $l_m$ attains the highest value for Edit queries, indicating the effectiveness of our activation loss. Then our memory loss can effectively push the activation values of Past-edit and Random queries to 0, thus disabling the patch on irrelevant examples. The

Table 6: The Success Rate (SR), Generalization Rate (GR), Edit Retain Rate (ER), Training Retain Rate (TrainR), Test Retain Rate (TestR) of Transformer-Patcher (T-Patcher) with a fixed memory set.

| Editor | FEVER Fact-Checking BERT-base (110M) | | | | | zsRE Question-Answering BART-base (139M) | | | | |
|---|---|---|---|---|---|---|---|---|---|---|
| | SR | GR | ER | TrainR | TestR | SR | GR | ER | TrainR | TestR |
| T-Patcher | 1.00 | 0.82 | 0.999 | 1.000 | 1.000 | 1.00 | 0.82 | 0.97 | 0.999 | 0.997 |

Table 7: The experimental results when utilizing all data in $D_{edit}$ as a single run of SME. E represents how many edits have been conducted. N represents how many mistakes have been made by the initial model $f_0$ on the entire edit set $D_{edit}$.

| Taks | SR | GR | ER | TrainR | TestR | E | N | Editor |
|---|---|---|---|---|---|---|---|---|
| **FEVER** | 1.00 | 0.82 | 1.00 | 0.999 | 1.000 | 998 | 1231 | T-Patcher |
| **zsRE** | 0.99 | 0.81 | 0.97 | 0.912 | 0.948 | 2308 | 2766 | |
| **FEVER** | 1.00 | 0.54 | 0.16 | 0.998 | 1.002 | 1250 | 1231 | FT(all)+KL |
| **zsRE** | 1.00 | 0.69 | 0.14 | 0.936 | 0.974 | 2821 | 2766 | |
| **FEVER** | 1.00 | 0.89 | 1.00 | 0.717 | 0.709 | 1588 | 1231 | SERA |
| **zsRE** | 1.00 | 0.90 | 0.97 | 0.728 | 0.694 | 3558 | 2766 | |

KL Patch has the lowest activation value of Edit query on both tasks, which explains the lower SR of QA in Table 3.

**Editing results of Transformer-Patcher with fixed memory set** The experimental results 1 are obtained using a memory set that is updated with the editing proceeds. Thus in Table 6 we present the editing results of Transformer-Patcher using a fixed memory set. We only observe a slight decline in ER and a slight rise in TrainR. The results further show the robustness of our method. Besides, we have to highlight that our method allows us to save more previous edits as memory and leverage more memories in the training process. Because we do not need to save original raw data but only corresponding input queries (several constant vectors that do not require gradients). On the contrary, KL requires feeding a mini-batch of raw data into the pre-edit model and post-edit model separately, thus the GPU memory becomes a restriction of the number of memories utilized in one batch. But Transformer-Patcher could apply hundreds of thousands of memory vectors in one batch and cost minimal GPU memory and computation resources.

**Contradictory of lower E and lower TestR in Table 2** It seems inconsistent that Transformer-Patcher has achieved fewer E and lower TestR than FT(all)+KL method. Because one would expect the model to reduce future errors and behave better on the test set by fixing errors. The phenomenon may be because of the data distribution gap between the edit set and the test set. Thus the improvement of "reducing future errors" can not directly lead to higher TestR. For FEVER, the accuracy of the initial model attains 88.3% on the edit set and 76.9% on the test set. For zsRE, the accuracy of the initial model attains 47.9% on the edit set and 23.1% on the test set. A distinct gap between the edit set and test set is observed. Thus we should comprehensively consider all metrics to evaluate methods. Another reasonable explanation is that our modified model may slightly overfit the edit example. But fitting more to edit examples may be a desired feature in actual applications because we expect the model to be closer to the real data met during deployment.

ACKNOWLEDGMENTS

This work was partially supported by the State Key Laboratory of Software Development Environment of China under Grant SKLSDE-2023ZX-16.

