# OpenReview forum: "Transformer-Patcher: One Mistake Worth One Neuron"
_ICLR.cc/2023/Conference — ICLR 2023 poster_

### Official Review · Reviewer_5q14 · 2022-10-24

**Confidence:** 3
**Correctness:** 3
**Technical Novelty And Significance:** 3
**Empirical Novelty And Significance:** 2
**Recommendation:** 6

**Clarity, Quality, Novelty And Reproducibility:**

The paper is clear about the method, with analyses of running metrics, patches, and generality. Proper metrics are designed for all aspects of performance is concerned. Model and training details are provided in the appendices, which is helpful to reproduce the results.

**Strength And Weaknesses:**

Strengths:

1. The paper presents a new problem to the community: How can a model editor fix a sequence of edits without forgetting? The problem is new and most of the previous methods can fail on this task.
2. Additionally, the paper proposes a few metrics to evaluate the capability of model editors in different dimensions: Reliability, generality, and locality. 5 metrics are proposed to evaluate those properties.
3. Compared to baseline models, Transformer Patcher shows effectiveness on all of these metrics, especially for GR and ER.
4. The proposed method can be lightweight for minor error correction since it avoids fine-tunings. It can also be scaled up to thousands of edits.

Weaknesses:

1. The problem proposed in the paper, i.e. sequential model editing, assumes the model would correct the errors one by one, while the problem can be re-formulated so that the model only performs one edit on a batch of errors. It might be practically useful for industrial applications where information may gradually accumulate and the product needs to be up-to-date, but it might not be the typical case.
2. The problem settings put other models in an unfavorable light. They're not designed to be operated multiple times, and running so many rounds of training to correct certain errors could lead to catastrophic forgetting (Shown in fig 5). In another word, they're not proper baselines to compare with.
3. Although the method can be scaled up to more edits, the cost is also significant. Since each edit will introduce additional parameters, the memory and computational overhead could be non-negligible, while the basic fine-tuning method does not introduce any additional parameters.

Questions:

Neurons are designed to be active only for certain errors, but dot-product seems not to be complex enough to recognize the patterns of errors. Thus, I'm interested in how often are the neurons activated. E.g. can they be activated with paraphrases? How often are they wrongly activated for irrelevant examples?

-----

After rebuttal: I decided to raise my score from 5 to 6 as I'm leaning to accept the paper.

**Summary Of The Paper:**

The paper presents a new problem, named Sequential Model Editing, that requires a model editor to make a sequence of error corrections while keeping all the previous edits, preserving model performance, as well as generalizing to equivalent inputs. Instead of modifying the model parameters, a set of new parameters (termed neurons in the paper) are added to the last layer of transformers. Because of the existence of the ReLU/GeLU activation functions, the neurons will not be activated for irrelevant inputs. Compared to previous models, Transformer Patcher is shown to be more effective on those metrics.

**Summary Of The Review:**

The paper proposes a new model editing problem: Sequential Model Editing, with a method named Transformer Patcher. The method is shown to be effective on the problem, but whether this is a useful task and whether baselines are properly configured need to be discussed.

---

> ### Author Response · Authors · 2022-11-15
> **Response to reviewer 5q14 (part1)**
>
> Thanks for your reviews, we hope the following comments could address your concerns.
>
> > The problem proposed in the paper, i.e. sequential model editing, assumes the model would correct the errors one by one, while the problem can be re-formulated so that the model only performs one edit on a batch of errors. It might be practically useful for industrial applications where information may gradually accumulate and the product needs to be up-to-date, but it might not be the typical case.
>
> 1. We believe that SME is an important question in the industry. For example, if an error occurs in an online medical QA model, it is too risky for us to wait and collect a batch of errors instead of fixing the error immediately. Therefore, to improve the user experience, we think in some situations we need to correct each mistake immediately. This led to our experimental setting: the instant one-by-one correction for a sequence of mistakes.
> 2. SME is important because most existing methods may struggle when forced to apply edits in a sequential manner rather than simultaneously [1], and the latest work about model editing [2] started to pay attention to the SME problem.
>
> > The problem settings put other models in an unfavorable light. They're not designed to be operated multiple times, and running so many rounds of training to correct certain errors could lead to catastrophic forgetting (Shown in fig 5). In another word, they're not proper baselines to compare with.
>
> Thanks for the review.
>
> We have supplemented our **concurrent work SERAC**[2] which is claimed to perform SME as a proper baseline in this revision.
>
> The SERAC[2] first employs a **scope classifier** to estimate if the input is relevant to (falls in the scope of) any **cached edit examples.** if so, it then employs a **counterfactual model (needs to have the identical output space as the original model)** to produce the output relying on **the most relevant cached example;** Otherwise, it returns the output of the original model. In our proposed SME experiment setting, the in-scope examples have the same label as the edit example, thus the function of **the counterfactual model** is to reproduce the answer of the relevant example. During the implementation of QA, we choose the Bart-base as the **counterfactual model**, but we find is not trivial for the Bart model to reproduce the answer, thus it is more practical to directly return the label of the **cached edit example**. We refer to this direct return method as **SERA** and supplement it as our baseline.
>
>    The experimental results are shown as follows, and relevant parts are modified in the revision:
>
>    For fact-checking,
>
>    | Editor    | SR   | GR   | ER   | TrainR | TestR |
>    | --------- | ---- | ---- | ---- | ------ | ----- |
>    | SERA      | 1.00 | 0.89 | 1.00 | 0.904  | 0.916 |
>    | T-Patcher | 1.00 | 0.82 | 1.00 | 0.999  | 1.000 |
>
>    For Question-Answering,
>
>    | Editor    | SR   | GR   | ER   | TrainR | TestR |
>    | --------- | ---- | ---- | ---- | ------ | ----- |
>    | SERA      | 1.00 | 0.90 | 0.98 | 0.906  | 0.901 |
>    | T-Patcher | 1.00 | 0.82 | 0.99 | 0.999  | 1.000 |
>
> According to experimental results, the SERA performs well in SR and ER, as it directly appends the edit example to a memory cache. It is observed that SERA achieves better generality (higher GR) but harms the model's overall performance (lower TranR and lower TestR) compared to the Transformer-Patcher. This may be because the trained scope classifier tends to activate the cached edit example given new inputs.

---

> ### Author Response · Authors · 2022-11-15
> **Respone to reviewer 5q14 (part2)**
>
> > Although the method can be scaled up to more edits, the cost is also significant. Since each edit will introduce additional parameters, the memory and computational overhead could be non-negligible, while the basic fine-tuning method does not introduce any additional parameters.
>
> As described on page 8, first paragraph, the model after thousands of edits only gets **1.4%** larger for Bert-base model and **4.5%** larger for Bart-base model than before. And this proportion will decrease if the method is applied to bigger models. We believe this cost is affordable for automatically correcting the model’s mistakes from time to time during deployment. In practice, we suggest using the transformer-patcher to provide a timely response for each mistake online, and after accumulating certain quantities of mistakes, we could fine-tune the original model on all accumulated mistakes, so that the patches can be removed. In this way, we could achieve a good balance between model size and editing effectiveness.
>
> > Neurons are designed to be active only for certain errors, but dot-product seems not to be complex enough to recognize the patterns of errors. Thus, I'm interested in how often are the neurons activated. E.g. can they be activated with paraphrases? How often are they wrongly activated for irrelevant examples?
>
> Thanks for the question.
>
> 1. **Can they be activated with paraphrases?** Yes, they can be activated by paraphrases, making Transformer-Pacher attain promising Generality Rates on two tasks (high GR), otherwise, the proposed Transformer-Patcher does not have generality.
> 2. **Activation Statistic Results** Then, we hope Table 4 (page 14) could address your concern about the activation situations of patches. Table 4 illustrates the average and deviation of absolute activation values of the patch when facing different examples. It is observed that the both average and deviation of activation values of irrelevant examples (e.g. previous edit examples, random examples from the testing set) are very close to 0 (0.05$\pm$0.1 for Fact-checking and 0.10$\pm$0.1 for Question-Answering), demonstrating that patches are seldomly activated by irrelevant examples. On the contrary, the activation value of the correspondent example is very high  (10.25$\pm$2.3 for Fact-checking and 6.78$\pm$2.58 for Question-Answering). Figure 6(c) reflects this as well.
>
> [1]  https://arxiv.org/pdf/2111.13654v1.pdf
>
> [2]  https://arxiv.org/abs/2206.06520

---

> ### Author Response · Authors · 2022-11-20
> **Request for response**
>
> Dear reviewer 5q14,
>
> The rebuttal phase 1 has ended, and we have provided detailed explanations and supplemented experiments to answer your questions. We wonder if it is possible for you to acknowledge us if our responses could address your concerns. We thank you for your time and your helpful suggestions.
>
> Best regards,
>
> Authors

---

> ### Author Response · Authors · 2022-12-02
> **Request for feedback**
>
> Dear reviewer 5q14,
>
> We have posted several comments to address your concerns. We have provided extra explanations to answer weaknesses 1,3 and your questions. We have supplemented the latest work: SERAC[1] about Model Editing as our baseline to address weakness 2. We wonder if our responses have addressed your concerns
>
> Look forward to your reply.
>
> Best regards,
>
> Authors
>
> [1] https://arxiv.org/abs/2206.06520

---

> ### Comment · Reviewer_5q14 · 2022-12-03
> **Response to the rebuttal**
>
> Thanks for the response and sorry for the late reply.
>
> For the usage of this method and the associated overhead:
> The patcher alone might not be sufficient for the actual applications. It makes sense that the patcher should be equipped with another fine-tuning-based method so that the growth of the model size is controllable. Thanks for pointing out the overhead of the method, it is reasonable, given the number of errors it corrects.
>
> For the generalizability: Thanks for providing more details. It would be nice if you could have at least a discussion in the final version of the paper.
>
> For the baselines: A stronger baseline makes the paper more convincing. The concurrent work you mentioned (I skimmed through the paper) is better than naively applying fine-tuning methods, but how about fine-tuning the model with all the errors so far? For example, upon the occurrence of the i-th error, the model would be fine-tuned with all the previous i errors, starting from the original model. I understand this comes with expensive costs, but that's something you may consider comparing with.
>
> As some of my concerns are addressed, I raise my score to acceptance.

---

> > ### Author Response · Authors · 2022-12-06
> > **Thanks for the reply to the rebuttal**
> >
> > Dear reviewer 5q14,
> >
> > Thanks for raising your score!
> >
> > According to your suggestions, we will try to add discussions about generalizability and supplement relevant fine-tuning-based baselines in the final version of this paper.
> >
> > Thanks for your valuable reviews for further improving our work.
> >
> > Best regards,
> >
> > Authors

---

### Official Review · Reviewer_n4JS · 2022-10-25

**Confidence:** 2
**Correctness:** 2
**Technical Novelty And Significance:** 2
**Empirical Novelty And Significance:** 2
**Recommendation:** 6

**Clarity, Quality, Novelty And Reproducibility:**

This paper is well-written and organized and the code is attached as supplementary material. But the novelty of this work is limited.

**Strength And Weaknesses:**

(+) The paper is clear and easy to follow.

(+) For the Model Editing (ME) problem proposed by Mitchell et al., 2022a, this paper proposes a sequential model editing task (SME) with more practical significance and application scenarios as mentioned in introduction part.

(+) And then the authors give a solution to the SME problem and conduct experiments on both classification and autoregressive generation tasks to prove its ability to edit the model up to thousand times continuously and outperform other methods.

(+) The code is attached as supplementary material and I think should be released if this paper is accepted.

(-) Although the authors claim to support multiple edit operations, they did not perform a stress testing on the model, similar to testing the effect of the model in extreme cases, such as thousands or millions of edit operations. The effect of the final model may deviate from what is described in the paper.


**Summary Of The Paper:**

The author first proposed the Sequential Model Editing task which requires fixing a series of mistakes for a pre-trained Transformer model by a model editor. And then the author introduced Transformer-Patcher which can be an effective model editor to fix a series of mistakes without serious performance drop.

**Summary Of The Review:**

For some technical details in the model, I didn't do much checking. I prefer to accept this paper and also need to look at the feedback of other reviewers before final deciding.

---

> ### Author Response · Authors · 2022-11-15
> **Response to reviewer n4JS**
>
> > Although the authors claim to support multiple edit operations, they did not perform a stress testing on the model, similar to testing the effect of the model in extreme cases, such as thousands or millions of edit operations. The effect of the final model may deviate from what is described in the paper.
>
> Thanks for the suggestion.
>
> Actually, in our *Scale up to thousands of edits* experiment, we have employed all data of the edit set to simulate one edit data stream, leading continuous edit times to 2308 for QA and 998 for Fact-checking.
>
> We have not tested our method on **millions of edits**. However, as mentioned on page8, top: In practice, we suggest using the transformer-patcher to provide a timely response for each mistake online, and after accumulating certain quantities of mistakes, we could fine-tune the original model on all accumulated mistakes, so that the patches can be removed.  We think the ability to scale up to thousands of edits could satisfy the requirements of the real application for a deployed online model.
>
> Please let us know what else we can do to help you to recommend our paper. Thanks for your time.

---

### Official Review · Reviewer_AwJg · 2022-10-26

**Confidence:** 4
**Correctness:** 2
**Technical Novelty And Significance:** 3
**Empirical Novelty And Significance:** 3
**Recommendation:** 6

**Clarity, Quality, Novelty And Reproducibility:**

[Clarity]
The writing is not easy to follow and may require multiple passes of reading. The figures introduce confusion.
1. It needs to be clarified what does Figure 1 like to highlight. Comparing the effect of batch editing and sequential editing on user experience may be a more intuitive purpose.
2. Figure 3 needs to be clarified. This figure looks not match the descriptions in Section 4. The layer-N block is part of the "patch" while it contains another patch. On the left side, the model prediction is an input to the patch. This looks misleading as well.
3. Eq10 has a redundant line with a_p=0
4. Figure 4 missed labels for the x and y-axis. 4(a)'s legend overlaps T-patcher's results.

[Novelty]
The method is novel.

[Reproducibility]
The paper needs to include a detailed description of its memory management mechanism. The description of the experiment and data preparation looks informative.


**Strength And Weaknesses:**

[Strengths]
1. The proposed method looks novel and can scale to thousands of edits. The result looks very good.
2. The sequential edit setting is not much explored by previous works.
3. The exp setting in Table 2 is meaningful in that it "employs authentic examples where the model makes mistakes".

[Weaknesses]
1. The method looks slow and requires significant computation (ex: 23.8 seconds per edit for QA).
2. The paper should have compared the closest work SERAC [1]. SERAC can do sequential model editing and make no changes to the original base model.
3. The clarity has a large room for improvement. Please see the next section for details.
4. The claim in the last paragraph of section 4 is not supported. The benefit of editing the last block can be supported by applying the same method to other layers. Besides, more aggressive paraphrasing of facts/questions could be applied to prove better generalizability or create more challenging cases like in [1].
5. The paper needs a detailed description of how it manages the memory to store previous examples. Does the method sensitive to the amount of stored data? What is the value of d_m used in the experiments?
6. The ablation in Table 3 can be more fine-grained. For example, ablating L_m1 and L_m2 separately.
7. To show SME is a setting that ME methods may fail, Table 1 could add the batched results of MEND/KE.

[1] https://arxiv.org/abs/2206.06520

**Summary Of The Paper:**

This work addresses the model editing problem where the edits are applied sequentially. The paper proposes to add one neuron per edit to the FFN layers of the last transformer module for language models. The hidden dimension of the FFN module is therefore increased by one after one edit is applied. The weights associated with the added neuron are optimized by Adam with the edit and some previous data stored in memory. The experiments evaluate the proposed method with MEND, KE, and some variants of fine-tuning under the sequential model editing setting. The results of two model editing datasets (FEVER, zsRE) show that the proposed method achieves a high edit success rate without affecting non-edit cases.

**Summary Of The Review:**

The paper provides a novel method for a less explored problem setting in model editing. The experiment results look promising. However, as mentioned above, the execution has many parts that need significant improvements to support the claims better. The mixed pros and cons lead to a borderline score.


[Update]

======== After rebuttal ========

I appreciated the author's detailed responses. Most of my concerns are addressed, although some still need to be fully satisfactory. I have exchanged opinions with other reviewers and agree that the current form's pros slightly out-weight the cons. The score was updated to reflect the overall evaluation. Below are some points that I hope the paper could further improve:
1. Add stronger support for generalizability.
2. Add a section to discuss the limitations. Large editing complexity could be one. Generalizability is another arguable one. I believe there are others, and I hope the author discloses all for the community to benefit from your findings.
3. A better comparison with prior work. The proposed SERA (simplified from SERAC) does not provide a convincing excuse to make the simplification of the SOTA method. Interestingly, SERA has better GR (generalizability) than the proposed method, making the standing of this work less solid. Anyway, SERA is still a nice baseline to add; therefore, I do not consider this a deal breaker.

---

> ### Author Response · Authors · 2022-11-15
> **Response to reviewer AwJg (part 1)**
>
> Thanks for your valuable reviews, we hope the following comments could address your concerns
>
> > The method looks slow and requires significant computation (ex: 23.8 seconds per edit for QA).
>
> The edit time of our proposed Transformer-Patcher could be ameliorated by:
>
> 1) **Using cache when training patches**: In the submission version, we compute all Transformer Layers for each edit. Because it is only the last layer where we add patches, the output of the previous layers could be cached and reused. We optimize the average edit time for QA to **18.9s** using the cache.
> 2) **Reducing memory examples**: we have conducted ablation studies about the number of memory examples (see below) and found that the performance is not sensitive to the number of memory examples. Therefore, we could boost the editing efficiency to **12.4s** by reducing memory examples to **10000 examples** (originally we use 40000 memory examples).
>
> We think this editing time and edit consumption is affordable for an online model to correct its behaviors from time to time. And the method is computationally efficient for inference of the online model because we do not involve any extra models but just a few added patches (ex, the Bert-base gets 1.5% larger after 998 edits and the Bart-base gets 4.5% larger after 2308 edits).
>
> We have revised the relevant part on page 7.
>
> >  The paper should have compared the closest work SERAC [1]. SERAC can do sequential model editing and make no changes to the original base model.
>
> Thanks for your valuable suggestions. The SERAC is our **concurrent work**. And we supplemented one variant of SERAC: **SERA** as our baseline in this revision.
>
> The SERAC first employs a **scope classifier** to estimate if the input is relevant to any **cached edit examples.** if so, it then employs a **counterfactual model which has the identical output space as the original model to produce the output relying on **the most relevant cached example;** Otherwise, it returns the output of the original model. In our proposed SME setting, the in-scope examples have the same label as the edit example, thus the function of **the counterfactual model** is to reproduce the answer of the relevant example. During the implementation of QA, we choose the Bart-base as the **counterfactual model**, but we find is not trivial for the Bart model to reproduce the answer, thus it is more practical to directly return the label of the **cached edit example**. We refer to this direct return method as **SERA** and supplement it as our baseline.
>
> The experimental results are shown as follows, and relevant parts are modified in the revision:
>
> For fact-checking,
>
> | Editor    | SR   | GR   | ER   | TrainR | TestR |
> | --------- | ---- | ---- | ---- | ------ | ----- |
> | SERA      | 1.00 | 0.89 | 1.00 | 0.904  | 0.916 |
> | T-Patcher | 1.00 | 0.82 | 1.00 | 0.999  | 1.000 |
>
> For Question-Answering,
>
> | Editor    | SR   | GR   | ER   | TrainR | TestR |
> | --------- | ---- | ---- | ---- | ------ | ----- |
> | SERA      | 1.00 | 0.90 | 0.98 | 0.906  | 0.901 |
> | T-Patcher | 1.00 | 0.82 | 0.99 | 0.999  | 1.000 |
>
> Experimental results show that SERA performs well in terms of SR, and ER, as it directly appends the edit example to a memory cache. SERA achieves better generality (higher GR) but harms the model's overall performance (lower TranR and lower TestR) compared to the Transformer-Patcher. This may be because the trained scope classifier tends to activate the cached edit examples given new inputs.
>
> In addition, we have also tested if SERA could scale up to thousands of edits. The experimental results are as follows:
>
> For fact-checking:
>
> | Editor    | SR   | GR   | ER   | TrainR | TestR |
> | --------- | ---- | ---- | ---- | ------ | ----- |
> | SERA      | 1.00 | 0.89 | 1.00 | 0.717  | 0.709 |
> | T-Patcher | 1.00 | 0.82 | 1.00 | 0.999  | 1.000 |
>
> For question-answering:
>
> | Editor    | SR   | GR   | ER   | TrainR | TestR |
> | --------- | ---- | ---- | ---- | ------ | ----- |
> | SERA      | 1.00 | 0.90 | 0.97 | 0.728  | 0.694 |
> | T-Patcher | 0.99 | 0.81 | 0.97 | 0.912  | 0.948 |
>
> We find that it is difficult for SERA to *preserve the model's overall performance (low TrainR and TestR)* when applying lots of edits.
>
> It is also worth mentioning that SERA and two hyper-net baselines (MEND and KE) require:
>
> 1. **extra training phase**: MEND and KE need to train a hyper-net and SERA needs to train the scope classifier,
> 2. **extra training data:** the in-scope examples (ex. the paraphrasing of the question) are required to train the hyper-net / scope classifier, while the in-scope examples are only used for evaluating our method.  Besides, these in-scope examples are barely provided in the training set of the original model in real industrial applications, thus training MEND/KE/SERA may need extra automatic techniques to generate the in-scope examples.
>
> Therefore, one strength of our method is that it is directly applicable without extra training phase and training data.

---

> ### Author Response · Authors · 2022-11-15
> **Response to reviewer AwJg (part2)**
>
> > [Clarity] The writing is not easy to follow and may require multiple passes of reading. The figures introduce confusion.
> >
> > 1. It needs to be clarified what does Figure 1 like to highlight. Comparing the effect of batch editing and sequential editing on user experience may be a more intuitive purpose.
> > 2. Figure 3 needs to be clarified. This figure looks not match the descriptions in Section 4. The layer-N block is part of the "patch" while it contains another patch. On the left side, the model prediction is an input to the patch. This looks misleading as well.
> > 3. Eq10 has a redundant line with a_p=0
> > 4. Figure 4 missed labels for the x and y-axis. 4(a)'s legend overlaps T-patcher's results.
>
> Thanks for your valuable comments.  We hope the revision described below could improve the clarity of this paper.
>
> 1. **Figure 1**: We want to illustrate our motivation to propose the SME problem: batched edit waits for accumulating a batch of errors, and possibly leaves the same error unfixed and bothers different users. As a result, instant correction is a better choice to improve the user experience.  We have revised the figure and highlighted our motivation in the caption of Figure 1.
> 2. **Figure 3**: Thanks for your valuable advice. The input and output of our model may indeed be misleading. We have revised this figure in the revision: 1) the left 'patch' has been corrected to an *operator* 'patcher' (add correspondent patches and train it), and the dashed line has been replaced by an arrow to express that the right part illustrates one patcher. We hope the revised figure 3 is more clear for you.
> 3. **Eq10**: Thanks for the suggestion, we have removed the redundant line.
> 4. **Figure 4**: For Figure 4a), the legend does not overlaps the T-Patcher. The T-Patcher stops earlier than MEND and KE because MEND and KE harm the model's overall performance and thus make more errors during editing.  As mentioned in the caption: "Different methods have different edit times, we plot until they converge."
>
> Please leave comments if there is something else, we are very glad to further improve the clarity of our paper.
>
> > The paper needs a detailed description of how it manages the memory to store previous examples. Does the method sensitive to the amount of stored data? What is the value of d_m used in the experiments?
>
> Thanks for your questions. We have explained how to construct the memory set in Appendix B of the submission. And we hope the comments below could address your concern.
>
> **Memory Management** Specifically, we have sampled 40000 examples from $D_{train}^{\prime}\setminus{D_{tr}}$, where $D_{train}^{\prime}$ is the training set of the original model,  where $D_{tr}$ is subsampled from $D_{train}^{\prime}$ to evaluate how the post-edit model performs on training data. Instead of caching raw examples, we just store all correspondent input vectors of the last FFN layer's input as the memory and reuse them while training patches. As fact-checking BERT-base model utilizes just the first token of input for classification, we only store the hidden state of the 1st token in each subsampled memory example, leading to $d_m=40,000$. As for Question-Answering, we store all token's hidden states in one memory example, leading to $d_m=201,715$. As the editing goes on, we **update the memory by appending the correspondent vectors** into the memory. The same memory management is employed for KL-based fine-tuning baselines.
>
> **Does the method sensitive to the number of stored memories?: No, the impact looks marginal**
>
> 1. First, we would like to mention that we have the experiment results about **not updating memory while editing** in Appendix C, Table 6 in the submission. The method is not sensitive to the **memory updating mechanism**, only a very slight of decrease of ER and an increase of TraniR and TestR are observed, which is reasonable.
>
> 2. Then, we have supplemented an ablation study about the number of memory examples on QA (using 5 fold of 20 folds as well):
>
>    | Memory Example Number | SR   | GR   | ER   | TrainR | TestR |
>    | --------------------- | ---- | ---- | ---- | ------ | ----- |
>    | 5000                  | 0.98 | 0.84 | 0.98 | 0.975  | 0.979 |
>    | 10000                 | 0.99 | 0.84 | 0.99 | 0.982  | 0.986 |
>    | 20000                 | 1.00 | 0.83 | 0.99 | 0.994  | 0.992 |
>    | 40000                 | 1.00 | 0.81 | 1.00 | 0.997  | 0.995 |
>
>    It is observed that our method is not very sensitive to the number of memory examples, reducing memory examples causes slight drops in SR, ER, TrainR, and TestR, and a  slight increase in GR.

---

> ### Author Response · Authors · 2022-11-15
> **Response to reviewer AwJg (part 3)**
>
> > The claim in the last paragraph of section 4 is not supported. The benefit of editing the last block can be supported by applying the same method to other layers. Besides, more aggressive paraphrasing of facts/questions could be applied to prove better generalizability or create more challenging cases like in [1].
>
> Thanks for your valuable suggestions, the last paragraph in section 4 may not be well-supported, thus we have supplemented an ablation study about how the position of the patch influences the editing performance. We have revised the paragraph and we hope this comment could address your concern.
>
> First, we are glad to share our considerations for patching the last FFN layer:
>
> 1. Patching the last layer ameliorates the editing efficiency since all computation results of previous layers could be cached and reused while editing;
> 2. In autoregressive generation tasks, neurons patched in the other layers could influence all tokens in one edit example owing to the following self-attention blocks, leading the patch should be optimized for simultaneously rectifying wrong tokens and retaining right tokens. While the patches added to the last layer only need to correct the wrong tokens, which may help the optimization.
> 3. Patching the last layer could potentially improve the generality of the method since previous studies found that high-level semantic features are encoded at the top layers.
>
> The experimental results of ablating position of the patching layer on the QA task (using 5 of 20 folds) are as follows:
>
> | Patching layer | SR   | GR   | ER   | TrainR | TestR |
> | -------------- | ---- | ---- | ---- | ------ | ----- |
> | 0              | -    | -    | -    | -      | -     |
> | 1              | -    | -    | -    | -      | -     |
> | 2              | 0.97 | 0.74 | 0.97 | 1.000  | 0.994 |
> | 3              | 0.98 | 0.76 | 0.98 | 0.997  | 0.998 |
> | 4              | 1.00 | 0.79 | 1.00 | 0.996  | 0.995 |
> | 5              | 1.00 | 0.81 | 1.00 | 0.997  | 0.995 |
>
> The observation is as follows:
>
> 1. First, patching the bottom layer (layer 0 and layer 1) can not make effect edits: this may be because patching the bottom layer severely influences every token in the input sequence, making optimization difficult. From layer 2. Transformer-Patcher could make effective edits.
> 2. Then, compared to other metrics, what the patching position influenced most is indeed GR, which increases from 0.74 of layer 2 to 0.81 of layer 5. We think this observation could support our claim.
>
> Besides, the paraphrasing of facts of [1] (e.g. True/False question, changing the subject of the question) may can not directly be employed in our experimental setting. Because in SERAC[1], the counterfactual model is trained to recognize these paraphrasing patterns and produce the right answer. However, in our experiments, neither the original model nor the added patch has seen these patterns.  But we do think creating more challenging cases for SME like [1] is one critical part of our future work.
>
> > The ablation in Table 3 can be more fine-grained. For example, ablating L_m1 and L_m2 separately.
>
> Thanks for your suggestion, we have ablated $l_{m_1}$ and $l_{m_2}$ separately in this revision.
>
> First, $l_{m_2}$ only distances the activation value of the edit example and memory examples, it does not assure that memory examples do not activate the patch, thus leading patch's failure. Experiments do show the model's overall performance is destroyed after about 50 edits.
>
> So we supplement Table 3 with experimental results of memory loss without $l_{m_2}$. The experiment results show that only $l_{m_2}$ could produce promising results, and adding $l_{m_2}$ could improve the performance a little on the basis of $l_{m_1}$.
>
> For Fact-Checking:
>
> | Patch         | SR   | GR   | ER   | TrainR | TestR |
> | ------------- | ---- | ---- | ---- | ------ | ----- |
> | w/o $l_{m_2}$ | 1.00 | 0.82 | 0.95 | 0.994  | 0.992 |
> | T-Patcher     | 1.00 | 0.82 | 1.00 | 0.999  | 1.000 |
>
> For Question-Answering:
>
> | Patch         | SR   | GR   | ER   | TrainR | TestR |
> | ------------- | ---- | ---- | ---- | ------ | ----- |
> | w/o $l_{m_2}$ | 0.95 | 0.82 | 0.94 | 0.991  | 0.984 |
> | T-Patcher     | 1.00 | 0.82 | 0.99 | 0.997  | 0.996 |

---

> ### Author Response · Authors · 2022-11-20
> **Request for response**
>
> Dear reviewer AwJg,
>
> The rebuttal phase 1 has ended, and we have provided extensive responses (including explanations and experiments) to answer your questions. We wonder if it is possible for you to acknowledge us if our responses properly address your concerns. We thank you for your time and your helpful suggestions.
>
> Best regards,
>
> Authors

---

> ### Author Response · Authors · 2022-12-02
> **Request for feedback**
>
> Dear reviewer AwJg,
>
> We have posted extra experiment results (e.g., a supplemented baseline and other ablation studies) and clarifications to answer your questions. We have also updated our paper according to your suggestions. We wonder if you can let us know whether our responses address your concerns.
>
> Look forward to your reply.
>
> Best regards,
>
> Authors

---

### Official Review · Reviewer_YxK1 · 2022-11-02

**Confidence:** 4
**Correctness:** 3
**Technical Novelty And Significance:** 3
**Empirical Novelty And Significance:** 4
**Recommendation:** 8

**Clarity, Quality, Novelty And Reproducibility:**

- Clarity: Great. The paper is clearly written and easy to follow. Some details and phrasing can be made more clear, but they do not harm any main points.
- Quality: Great. Clearly designed and explained task, goal, method, metrics, and results. The task is interesting and of practical value. The goal and method makes sense and are clearly described. The method is simple yet effective. The metrics are carefully chosen to reflect different aspects of the desiderata. The experimental support is strong and definitive that MSE poses a new challenge and the proposed Transformer-Patcher works much better than previous methods.
- Novelty: Great. Sequential Model Editing is a novel, interesting and practical setting that poses a different challenge than the previous studies on Model Editing. This is one step further towards the realistic scenario of the online updating of ML models. The results shown previous models which are good at single edits are not necessarily good at continuous editing. The model design is simple yet effective. The loss design shows the authors' consideration for the task.
- Reproducibility: Good. Experiment details described in the paper and the appendix. Code is provided.


**Strength And Weaknesses:**

### Strengths

1. The proposed task of Sequential Model Editing (SME) is novel, interesting and of practical value. With the increasing adoption of large language models, training, and even fine-tuning, can become computationally expensive or inefficient. We've seen rising interest in Model Editing where people try to find efficient ways to fix undesired model behavior. However, existing works focus on single mistakes from an input-output example. In real ML applications, a model needs to take user requests constantly, any downtime or unfixed mistakes could harm user experience. The SME setting manifests the need for a model-editing method that is timely (quick to apply), efficient (cost/computation effective) and effective (fixes the mistake) even after a series of accumulating edits (not just a single edit).
2. The SME desiderata (Sec 3), namely reliability, generality, and locality, are clearly stated and formulated. It gives a clear base for what the authors are looking for.
3. The design of Transformer-Patcher (Sec 4) is simple and of intuitive sense.
4. The design of the objective losses (Sec 4.2) are clearly explained and are of intuitive sense. In Sec 5.3, the ablation study suggests the usefulness of the losses. I do think the exact formula for l_{m1} (Eq (17), see Q1) seems a bit overcomplicated, but this is relatively minor.
5. The experimental support is strong that SME is an important and challenging setting and that Transformer-Patcher clearly outperforms competitive baselines. This strength can be further break down to
   1. The 5 metrics (Sec 5.1) are carefully chosen to reflect different aspects namely reliability, generality, and locality, and are clearly described.
   2. Competitive baselines are chosen (Sec 5.1), including several variants of fine-tuning and recently proposed HyperNetworks KE (Cao et al., 2021) and MEND (Mitchell et al., 2022a).
   2. Results (Table 1 and 2) show almost unanimous improvements over baselines across all metrics and both tasks, often by a big margin.
   3. Main results (Table 1) and metric progression over number of edits (Figure 4 and 5) show that HyperNetworks completely fail SME, despite being successful in single step ME.
   4. Ablation study (Table 3, Figure 6) shows that their proposed losses are effective in preventing regression across edits.
7. The paper is clearly written and easy to follow. Some details and phrasing can be made more clear, but they do not harm the main strengths.
8. Experimental details are provided in the appendix. Code is provided in the supplementary material. Both are good for reproducibility,

### Weaknesses

Nothing major. See my Questions and LPs. I think clarifying them can further improve the paper's strength and help me gain more confidence in recommending the work.

### Questions for the authors
1. P5, Eq (17), "S(v; k) = Avg[TopK(exp(v); k)]". Why do we need TopK? If we care about maximum ("Although constraint 15 is about the maximum..."), why not simply take the maximum, or use a surrogate like softmax?
2. Table 1 and 2. Why are TrainR and TextR generally < 1? If we are fixing errors (SR↑, GR↑) and not forgetting previous fixes (ER↑), then we should see the final models be more accurate than the initial models (TrainR>1, TestR↑>1)? The authors argue at one point (P8, L6) that there can be distribution shit between the train and test set. However, we are also seeing TrainR < 1 for all methods and datasets. Are we missing something here?

### Localized points (LPs)
1. P3, Eq (4), Locality, "$\forall x_j \in I_{x_e}, f′(x_j) = y_{x_j}$". $\forall x_j \in I_{x_e}, f′(x_j) = f(y_{x_j})$ makes sense as "locality" to me. The former means that the edited model $f'$ should give correct predictions to irrelevant examples, which is the learning task itself. The latter means that the edited model $f'$ should preserve the same prediction as the unedited model $f$ to irrelevant examples, which is more aligned with "locality" -- we do not want to change the model behavior at irrelevant examples, because we believe the model is mostly correct but cannot be sure without testing it against the ground-truth data.
2. P7, L3 after the figures, "The ideal edit length for each run is about 63 for
FC and 139 for zsRE." What is "edit length" and what does "ideal'' mean here? Is edit length the size of the D_edit or the number of edits the proposed method performed? Is it a hyperparameter that the authors decided based on experimental observations? Does "ideal" mean that the proposed method performs worse after 63/139 mistakes or edits? It seems like this claim belongs to results, not the settings.
2. P7, Experiment Details. It's worth mentioning the size of the test datasets. How many examples are for the test for each task? What is the total number of time steps T for each of the 20-fold runs?
3. P7, Table 1, "* denotes that SR of T-patcher on QA is not 1 but very close to 1." Does it mean other SRs in the column are actually exact 1? I suppose that the authors mean that the T-patcher SR is >0.995 so it rounds to 1.00 if we only keep two decimal points, but I cannot be sure. If that is indeed the case, I suggest improving the phrasing, e.g. sampling write out the exact number like "* T-patcher SR on QA = 0.9965".
4. P7, bottom, "Table 1 shows that Transformer-Patcher achieves good performance for about one hundred edits, ..." Exactly *how many* edits? And again, does "edit" here mean the size of the test set D_edit, or the number of times the proposed method applied a patch?

### Typography/Minor points
1. Abstract, "... either fail to make a sequence of edits or to remember previous edits" -> "... either fail to make a sequence of edits or fail to remember previous edits".
2. P2, top, "Therefore, an ideal model editor should conduct ..." Perhaps a better phrasing can be "Therefore, an ideal model editor should enable _countinuous_ fixing of newly emerged mistakes in a both effective and efficient manner."
3. P2, middle, "To handle SME.We introduce ..." -> "To handle SME, we introduce ..."
4. P3, Figure 2. Unify the notations of $y_1$ or $y_{x_1}$.
5. P3, Sec 3. I suggest use $y_e$, $y_j$, $y_1$, etc instead of $y_{x_e}$,  $y_{x_j}$, $y_{x_1}$, etc to reduce clutter.
6. P3, Property 1-3. Start with either capitalized or uncapitalized letters.
7. P3, Eq (5), "$f_t(x_k) = y_{x_k} , \{k | f_{k-1}...$" -> "$f_t(x_k) = y_{x_k}$ , for $k$ where $f_{k-1} ...$".
8. P4, Figure 3. Maybe revise the UK Prime Minister example...
9. P6, Eq (21). Is it $T \sum N_t$ or $T N_t$? It makes sense to me if you first average over N_t samples at time t and then average the average over T time steps.
10. P6, Eq (23), "we compare the performance the final model of f_T and the initial model f_0 on sub-training test D_tr." -> "we compare the performance of the final model of f_T and the initial model f_0 on the subsampled training set D_tr".
11. P6, Eq (20-22). I recommend moving the denominators before the \sum's. It would look more intuitive to me.


**Summary Of The Paper:**

The authors introduce the task of Sequential Model Editing (SME) where a model needs to adapt to and fix its mistakes from a stream of incoming input-output examples of a certain machine learning task. The setting extends Model Editing (ME), where people focus on editing the model to fix a single occurrence of an observed mistake. The authors define the desired properties of SME, namely reliability, generality, and locality. The authors then propose Transformer-Patcher for SME, which adds a single neuron to the last fully-connected layer of a Transformer; along with specially designed losses and training schemes for Transformer-Patcher. The authors measure 5 metrics that cover all the three desired aspects. Experiments show that 1) HyperNetwork methods fail MSE, 2) the proposed Transformer-Patcher clearly outperforms the fine-tuning and HyperNetwork baselines, often by a big margin, 3) Transformer-Patcher is effective up to thousands of edits. The authors further provide analysis on the metrics progression of the tested methods, and ablation study of the losses.

**Summary Of The Review:**

This is a strong work that extends an existing task (Model Editing) to a more realistic and challenging setting (Sequential Model Editing), where previous successful methods fail. The authors clearly state the desiderata, method, and metrics, along with the intuitions behind them. Experimental results show strong support that the proposed Transformer-Patcher is effective by a clear margin. Further analysis confirms the effectiveness of the design. Model Editing is of interest as the language models grow ever large and are increasingly adopted in real applications. I think this is a work that would interest audiences from the application of large language models, model editing, model adaptation, continuous learning, adversarials and robustness, model security and more.

---

> ### Author Response · Authors · 2022-11-15
> **Response to reviewer YxK1 (part1)**
>
> Thanks for recognizing the value of our work, your comments are highly aligned with our paper. We have adjusted our paper in this revision according to your suggestions. And we hope the following comments could answer your questions.
>
> > P5, Eq (17), "S(v; k) = Avg[TopK(exp(v); k)]". Why do we need TopK? If we care about maximum ("Although constraint 15 is about the maximum..."), why not simply take the maximum, or use a surrogate like softmax?
>
> Thanks for the question. We empirically find that it is more efficient to involve TopK instead of only considering the maximum. In the preliminary experiments, we observed that taking only maximum makes memory loss more difficult to converge. This is reasonable because every time the patch is optimized just uses one memory vector if using Top1.
>
> > Table 1 and 2. Why are TrainR and TextR generally < 1? If we are fixing errors (SR↑, GR↑) and not forgetting previous fixes (ER↑), then we should see the final models be more accurate than the initial models (TrainR>1, TestR↑>1). The authors argue at one point (P8, L6) that there can be a distribution shift between the train and test set. However, we are also seeing TrainR < 1 for all methods and datasets. Are we missing something here?
>
> For our method, SR=1 and ER~=1 show that our method is capable to fix errors and retain the fixing effect. However, this does not assure that the model attains higher accuracy on its training set and testing set. Besides, the distribution shift in the training set (for TrainR), test set (for TestR), and edit set (for SR, GR, and ER) is one reason. The method could be trained to overfit the edit set, thus may lead TranR and TestR to smaller than 1. However, fitting more to edit examples is not that worrying in the real application, because we expect the model to be closer to the real data met during deployment.
>
> More detailed information could be found on Page 15, last paragraph.
>
> > P3, Eq (4), Locality, "∀xj∈Ixe,f′(xj)=yxj". ∀xj∈Ixe,f′(xj)=f(yxj) makes sense as "locality" to me. The former means that the edited model f′ should give correct predictions to irrelevant examples, which is the learning task itself. The latter means that the edited model f′ should preserve the same prediction as the unedited model f to irrelevant examples, which is more aligned with "locality" -- we do not want to change the model behavior at irrelevant examples, because we believe the model is mostly correct but cannot be sure without testing it against the ground-truth data.
>
> Thanks for the question, we think both two definitions make sense.
>
> Your proposed definition is reasonable as it strictly concentrates on whether the model changes its behavior, while our definition focuses more on the model's actual performance. Besides, our definition is a bit more friendly to the experimental implementation, because updating and recording the output of every pre-edit model requires extra time, computation, and memory space.
>
> More discussions are welcomed about which definition is more appropriate.

---

> ### Author Response · Authors · 2022-11-15
> **Response to reviewer YxK1 (part2)**
>
> > P7, L3 after the figures, "The ideal edit length for each run is about 63 for FC and 139 for zsRE." What is "edit length" and what does "ideal'' mean here? Is edit length the size of the D_edit or the number of edits the proposed method performed? Is it a hyperparameter that the authors decided based on experimental observations? Does "ideal" mean that the proposed method performs worse after 63/139 mistakes or edits? It seems like this claim belongs to results, not the settings.
>
> We evaluate the original model on the edit set so that we could find the mistakes of the original model, and we refer to this mistake number as the "ideal edit length": how many edits will be conducted if the model editor does not cause any errors while editing. Because we fix the size of the edit set, every model editor will have different edit times according to their performance. We have revised this part in this revision to make it more clear.
>
> > P7, Experiment Details. It's worth mentioning the size of the test datasets. How many examples are for the test for each task? What is the total number of time steps T for each of the 20-fold runs?
>
> Owing to the space limit, we put all this information in Appendix B, Experimental Details.
>
> > P7, Table 1, "* denotes that SR of T-patcher on QA is not 1 but very close to 1." Does it mean other SRs in the column are actually exactly 1? I suppose that the authors mean that the T-patcher SR is >0.995 so it rounds to 1.00 if we only keep two decimal points, but I cannot be sure. If that is indeed the case, I suggest improving the phrasing, e.g. sampling write out the exact number like "* T-patcher SR on QA = 0.9965".
>
> Yes, your understanding is correct, we have adjusted the caption of the table following your suggestion
>
> > P7, bottom, "Table 1 shows that Transformer-Patcher achieves good performance for about one hundred edits, ..." Exactly *how many* edits? And again, does "edit" here mean the size of the test set D_edit, or the number of times the proposed method applied a patch?
>
> About 140 edits for QA and 60 edits for Fact-checking on average. The editor only edits the model when encountering a mistake from the edit set. Thus the edits here refer to the number of times the proposed method edits the original model. It does not mean the number of times the proposed method applied a patch, because T-Patcher may apply multiple patches for one error of the QA task.
>
> Thanks for your question, we have revised this part to make it more clear.
>
> > **Typography/Minor points**
>
> Thanks for your valuable suggestions. We have corrected all mentioned minor points according to your advice.
>
> Thanks for recommending our paper, please leave comments if you still have concerns.

---

### Author Response · Authors · 2022-11-15
**General Comment**

We sincerely appreciate all reviewers' time and efforts in reviewing our paper. We are glad to find that reviewers generally recognize our strengths:

- Motivated by the fact that deployed online models may need timely and continuous automatic repair, we extend the existing Model Editing and proposed Sequential Model Editing task, which is **novel** [5q14,n4JS, AwJg,yXk1] and of **practical value** [n4JS, YxK1]
- We proposed 5 proper evaluation metrics to reflect the performance of one model editor from different aspects (Reliability, generality, and locality) [YxK1, 5q14]
- The proposed method: Transformer-Patcher is **novel** [AwJg], **effective** [YxK1, AwJg, n4JS, 5q14], and can **scale up to thousands of edits** [n4JS, 5q14, AwJg]

We thank the useful suggestions from the reviewers, which help a lot in further improvement of this paper. In addition to the pointwise responses below, major revisions for rebuttal are summarized as follows (marked with the red color in this revision):

**Localized points**

We have revised some localized points (e.g., Figure 1 and Figure 3 in the Introduction, equations, and other claims in our paper) following the suggestions of reviewer YxK1, AwJg. We sincerely appreciate the useful suggestions. And more suggestions are welcomed to improve the clarity of this paper.

**Ablation Studies**

We have supplemented several ablation studies to prove the robustness of our method and to support our claims in the paper.

- **Memory example number** As reviewer AwJg is curious about if our proposed method is sensitive to memory number, we have conducted an ablation study about the number of memory examples on the QA task.

  | Memory Example Number | SR   | GR   | ER   | TrainR | TestR |
  | --------------------- | ---- | ---- | ---- | ------ | ----- |
  | 5000                  | 0.98 | 0.84 | 0.98 | 0.975  | 0.979 |
  | 10000                 | 0.99 | 0.84 | 0.99 | 0.982  | 0.986 |
  | 20000                 | 1.00 | 0.83 | 0.99 | 0.994  | 0.992 |
  | 40000                 | 1.00 | 0.81 | 1.00 | 0.997  | 0.995 |

  The ablation results demonstrate that our method is **not sensitive** to the number of memories. Reducing the memory example number only leads to a slight drop in the method's performance, further validating the robustness of the Transformer-Patcher. We supplement the results in this revision.

- **Patching Layer Position** Following reviews of reviewer AwJg, we have conducted ablation experiments on which layer to add the patches. Concentrating on the QA task, we add patches on layer 0 to layer 5 of the Bart decoder respectively, the experimental results are as follows:

  | Patching layer | SR   | GR   | ER   | TrainR | TestR |
  | -------------- | ---- | ---- | ---- | ------ | ----- |
  | 0              | -    | -    | -    | -      | -     |
  | 1              | -    | -    | -    | -      | -     |
  | 2              | 0.97 | 0.74 | 0.97 | 1.000  | 0.994 |
  | 3              | 0.98 | 0.76 | 0.98 | 0.997  | 0.998 |
  | 4              | 1.00 | 0.79 | 1.00 | 0.996  | 0.995 |
  | 5              | 1.00 | 0.81 | 1.00 | 0.997  | 0.995 |

  It is observed that:

  1. Patching bottom layers (layer 0 and layer 1) could not make desirable edits.
  2. And what the patching position influence most is the generality of the Transformer-Patcher.

  We supplement the results in this revision.

Please contact us if we can do something else to help you better understand and recommend our paper.

---

### Author Response · Authors · 2022-11-17
**Request for Discussion**

Dear reviewers,

Thank you again for your valuable time and insightful comments.

Given that the author-reviewer discussion period is coming to a close soon, we request reviewers to let us know if our responses have resolved their concerns and if there are any other questions that we can address to help recommend our paper.

Best regards!

Authors

---

### Decision · Program_Chairs · 2023-01-20

**Decision:**

Accept: poster

**Justification For Why Not Higher Score:**

Comparison against baselines may not be super fair as previously baselines were tuned for a batched setting. Moreover, there were legitimate concerns about how this method would scale to realistic scenarios.

**Justification For Why Not Lower Score:**

The method is novel and moreover targets a new online knowledge acquisition setting.

**Metareview: Summary, Strengths And Weaknesses:**

This paper proposes an approach to editing (or patching) Transformers by adding neurons that encode the additional knowledge. The strengths of the paper are its consideration of a novel scenario and empirical performance. The weaknesses of the paper are scalability and potential unrealistic-ness of the one-by-one sequential editing scenario (as opposed to batched updates).

**Note From Pc:**

if the above contains the word "oral" or "spotlight" please see: "oral" presentation means -> notable-top-5% and "spotlight" means -> notable-top-25%. As stated in our emails, we are disassociating presentation type from AC recommendations

**Summary Of Ac-Reviewer Meeting:**

Some points that were mentioned in the discussion were:
- lack of comparison against methods such as SERAC may be okay since this paper targets an online knowledge acquisition setting unlike previous works.
- the novelty of the method
- importance of the problem
- whether this would scale to realistic settings